# URRL-IMVC: Unified and Robust Representation Learning for Incomplete Multi-View Clustering

## Abstract

Incomplete multi-view clustering (IMVC) aims to cluster multi-view data that are only partially available. This poses two main challenges: effectively leveraging multi-view information and mitigating the impact of missing views. Prevailing solutions employ cross-view contrastive learning and missing view recovery techniques respectively. However, they either neglect valuable complementary information by focusing only on consensus between views or provide unreliable recovered views due to the absence of supervision. To address these limitations, we propose a novel Unified and Robust Representation Learning for Incomplete Multi-View Clustering (URRL-IMVC). URRL-IMVC learns a unified embedding that is robust to view missing conditions by integrating information from multiple views and neighboring samples. Firstly, to overcome the limitations of cross-view contrastive learning, URRL-IMVC incorporates an attention-based auto-encoder framework to fuse multi-view information and generate unified embeddings. Secondly, URRL-IMVC directly enhances the robustness of the unified embedding against view-missing conditions through KNN imputation and data augmentation techniques, eliminating the need for explicit missing view recovery. Finally, incremental improvements are introduced to further enhance the overall performance, such as adaptive masking, dynamic initialization, etc. We extensively evaluate the proposed URRL-IMVC framework on various benchmark datasets, demonstrating its state-of-the-art performance. Furthermore, comprehensive ablation studies are performed to validate the effectiveness of our design.

## 1 Introduction

Multi-view data (Fu et al., 2020) is commonly collected and utilized in various domains, making multi-view clustering (MVC) a crucial tool for analyzing such data and uncovering its underlying structures (Chao et al., 2021; Chen et al., 2022). Previous research has proposed several approaches (Xu et al., 2022; 2021) achieving promising performance by exploiting consensus or complementary information between views. However, in real-world applications, some views may be partially unavailable due to sensor malfunctions or other practical reasons. Existing MVC methods heavily rely on complete views to learn a comprehensive representation for clustering, making them inadequate under such conditions. To address this issue, Incomplete Multi-view Clustering (IMVC) methods have been introduced to reduce the impact of missing views (Wen et al., 2023). Various IMVC approaches have been proposed, including matrix decomposition (Li et al., 2014), kernel-based (Liu et al., 2017), and graph-based (Gao et al., 2016) methods. With the superior feature representation ability demonstrated by deep learning, some IMVC methods have integrated deep learning techniques, known as Deep Incomplete Multi-view Clustering (DIMVC) methods, which we will mainly discuss below. The key challenges in the IMVC task revolve around two problems: i) effectively utilizing multi-view information, and ii) mitigating the impact of missing views. Previous DIMVC works (Wang et al., 2018; Lin et al., 2021; 2023; Jin et al., 2023; Liu et al., 2023) have employed two mainstream strategies to address these problems: 1) cross-view contrastive learning, and 2) missing view recovery. However, these strategies have inherent drawbacks.

A general framework for cross-view contrastive learning is illustrated in Fig 1a, which originates from the MVC approaches. In this framework, Deep Neural Network (DNN) auto-encoders are em-

ployed to extract embeddings for each view. The embeddings are then aligned using a contrastive loss, aiming to minimize the distance between embeddings from the same sample across different views while simultaneously maximizing the distance with other samples (Jin et al., 2023; Lin et al., 2021; Yang et al., 2023b). However, this framework primarily focuses on extracting consensus information in multi-view data, overlooking the valuable complementary information present. Additionally, the efficiency of the pair-wise contrastive strategy suffers as the number of views increases, and the effectiveness of this strategy diminishes due to less overlapped information between views (See Table 7 for experimental analysis). Theoretical analysis by Trosten et al. (2023) supports these observations, highlighting that contrastive alignment can reduce the number of separable clusters in the representation space, with this effect worsening as the number of views increases.

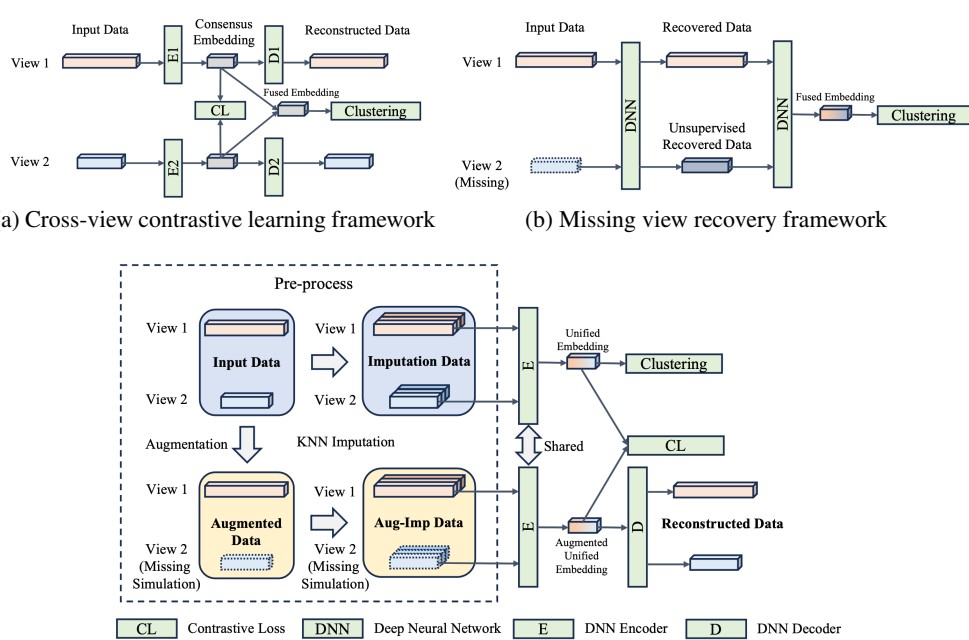

(a) Cross-view contrastive learning framework    (b) Missing view recovery framework

(c) Our unified learning framework

Figure 1: A comparison between our unified learning framework and commonly used cross-view contrastive learning and missing view recovery framework. The key difference lies in how the unified embedding for clustering is obtained. Our design 1c directly fuses multi-view information and utilizes KNN imputation and data augmentation to obtain unified and robust embedding under view-missing conditions, avoiding the drawbacks of 1a and 1b.

The missing view recovery framework, as depicted in 1b, is commonly adopted in IMVC approaches. Typically, a DNN is employed to recover the missing view, either in the data or latent space. Subsequently, MVC methods or another view fusion network are utilized for clustering based on the recovered views. However, the reliability of the recovered views is a concern since the recovery ability of DNNs relies on unsupervised training. Meanwhile in some instances, (Liu et al., 2023) for example, missing views are recovered by a fused embedding in the first stage, and subsequently used to generate another fused embedding for clustering in the second stage, introducing unnecessary complexity and inefficiency to the pipeline. We propose that a well-designed recovery-free method can achieve comparable performance to recovery-based methods while offering the advantages of simplicity and reduced computational overhead (See Table 12 for experimental analysis).

To address the aforementioned challenges, we propose a Unified and Robust Representation Learning framework for Incomplete Multi-View Clustering (URRL-IMVC). Our framework, depicted in Fig 1c, is designed to be cross-view contrastive learning-free and missing view recovery-free. First, in order to overcome the limitations of cross-view contrastive learning, our approach focuses on learning a unified embedding that captures the comprehensive representation. We achieve this by employing a scalable attention-based auto-encoder network, which intelligently fuses the infor-

mation from multiple views to generate the desired representation. Second, to tackle the issue of missing views, we aim to directly enhance the robustness of the unified embedding against view-missing conditions without explicitly recovering the missing views. We introduce two strategies to achieve this robustness. 1) We treat view missing as a form of noise and draw inspiration from successful applications of denoising and masked auto-encoders (Vincent et al., 2008; He et al., 2022). Our proposed approach randomly drops out existing views as a form of data augmentation to simulate the view missing condition. By reconstructing denoised input data from the unified embedding and imposing constraints between the augmented and un-augmented embeddings, we enhance the robustness of the unified representation. 2) As the old saying goes, "One cannot make bricks without straw", it is hard to directly learn to reconstruct a dropped-out view. We introduce k-nearest neighbors (KNN) as additional inputs, with a cross-view imputation strategy to fill in the missing or dropped-out views, providing valuable hints for reconstruction. We would like to highlight that while previous methods have focused on either fusing multi-view information (Wang et al., 2021; Lin et al., 2022) or incorporating neighborhood information (Nguyen et al., 2021; Wang et al., 2019b; Yang et al., 2020; Tu et al., 2021) for clustering, our approach represents one of the initial endeavors to fuse both aspects, thanks to the scalability of our proposed framework. Finally, we conduct experiments based on this framework and make incremental improvements to enhance clustering performance and stability. Some of the key enhancements include adaptive masking and positional encoding in the Transformer-based Encoder to filter out noise and emphasize critical information, as well as dynamic initialization in the Clustering Module to improve clustering stability.

To summarize, our main contributions are:

- We propose a unified representation learning framework that efficiently fuses both multi-view and neighborhood information, allowing for better capturing of consensus and complementary information while avoiding the limitations of cross-view contrastive learning.

- We proposed novel strategies, including KNN imputation and data augmentation, to directly learn a robust representation capable of handling view-missing conditions without explicit missing view recovery.

- Multiple incremental improvements and thorough ablation studies are conducted, leading to enhanced clustering performance and providing valuable insights for future research in this field.

- Through comprehensive experiments on diverse benchmark datasets, we demonstrate the superior performance of our unified representation learning framework, establishing it as the state-of-the-art method for incomplete multi-view clustering.

## 2 RELATED WORKS

Related traditional IMVC approaches are described in **Appendix A**. Deep neural networks (DNNs) have shown good performance in learning feature representation, which is beneficial for the IMVC task. Various IMVC approaches have integrated DNNs into their framework, denoted as DIMVC approaches. In terms of network architecture, DIMVC approaches can be divided into four categories. (1) Auto-encoder-based approaches (Lin et al., 2022; Jin et al., 2023; Lin et al., 2021; 2023). These approaches utilize auto-encoders to extract high-level features of each view, which are usually combined with contrastive learning or cross-view prediction to handle the incompleteness problem. (2) Generative network-based approaches. For the IMVC task, an intuitive solution is to complete the missing views with generative models, transforming it into an MVC task. Adversarial learning (Goodfellow et al., 2014) is commonly adopted by generative IMVC approaches including AIMVC Xu et al. (2019), PMVC-CG Wang et al. (2018), and GP-MVC Wang et al. (2021) to improve data distribution learning in the context of IMVC. (3) Graph Neural Network-based (GNN-based) approaches (Wang et al., 2022; 2018). These approaches aim to learn consensus representations from the structure information contained in the graphs constructed for each view. (4) Transformer (Vaswani et al., 2017) or attention-based approaches. The Transformer network has gained attention in recent years due to its successful application in various domains. Its architecture, along with its Multi-head Attention mechanism, has been particularly effective in capturing complex relationships. In the field of DIMVC, RecFormer (Liu et al., 2023) proposed a Transformer auto-encoder with a mask to recover missing views, while MCAC (Zhang & Zhu, 2023) and IMVC-PBI (Li et al., 2023a)

incorporated attention mechanisms into their frameworks. In this paper, we leverage an auto-encoder architecture based on the Transformer framework to address the challenges of the IMVC task.

## 3 THE PROPOSED METHOD

**Notations.** An incomplete multi-view dataset with $N$ samples and $V$ views is denoted as $X = \{X^{(1)}, X^{(2)}, \cdots, X^{(V)}\}$, $X^{(v)} \in \mathbb{R}^{N \times d_v}$, where $d_v$ denotes the dimension of $v$-th view. The view missing condition can be described by a binary missing indicator matrix $M \in \{0, 1\}^{N \times V}$, where $M_{ij} = 0$ indicates the $j$-th view of the $i$-th sample is missing and $M_{ij} = 1$ just the opposite. An extra restriction is imposed: $\sum_j M_{ij} \geq 1$, ensuring that at least one view is available for each sample, which is essential for the clustering task.

### 3.1 FRAMEWORK

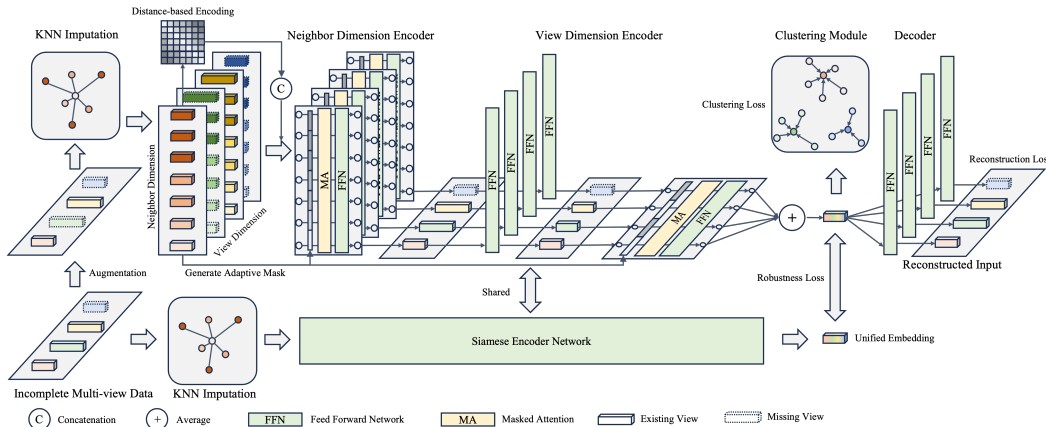

Figure 2: The framework of URRL-IMVC. During training, the input data is augmented to simulate view-missing conditions, and KNN Imputation provides hints for missing views, forming an input batch with both neighbor and view dimensions. This batch is fed into the auto-encoder network, consisting of the Encoder (including the Neighbor Dimensional Encoder and View Dimensional Encoder), the Decoder, and the Clustering Module. The Encoders fuse information from the neighbor and view dimensions to generate a unified embedding. The Decoder reconstructs the augmented input, and the Clustering Module produces clustering results. Additionally, an un-augmented embedding is obtained by passing the original input data through the shared Encoders. Three loss functions, including Reconstruction loss, Robustness loss, and Clustering loss, enhance robustness against view-missing conditions and encourage learning clustering-friendly embeddings.

Unlike many prior approaches in the field of MVC that employ view-specific auto-encoders for each view, we propose a novel framework using a unified auto-encoder that effectively fuses multi-view data. The network architecture, depicted in Fig 2, consists of three key modules: the Encoder $f$, the Decoder $g$, and the Clustering Module $h$. The Encoder can be further divided into two submodules: the Neighbor Dimensional Encoder (NDE) and the View Dimensional Encoder (VDE). To provide a formal description, the framework operates as follows. Given an incomplete multi-view data sample $\boldsymbol{x} = \{\boldsymbol{x}^{(1)}, \boldsymbol{x}^{(2)}, \cdots, \boldsymbol{x}^{(V)}\}$, $\boldsymbol{x}^{(v)} \in \mathbb{R}^{d_v}$ from dataset $X$ with its missing indicator vector $\boldsymbol{m} \in \{0, 1\}^V$, we apply K-Nearest-Neighbor (KNN) Imputation and Data Augmentation (KIDA), as described in section 3.2, to obtain the input for the auto-encoder network,

$$\bar{\boldsymbol{x}}, \bar{\boldsymbol{x}}'', \bar{\boldsymbol{m}}, \bar{\boldsymbol{m}}' = KIDA(\boldsymbol{x}, \boldsymbol{m}); \ \bar{\boldsymbol{x}}^{(v)}, \bar{\boldsymbol{x}}''^{(v)} \in \mathbb{R}^{k \times d_v}; \ \bar{\boldsymbol{m}}, \bar{\boldsymbol{m}}' \in \{0, 1\}^{k \times V} \tag{1}$$

where $\bar{\boldsymbol{x}}, \bar{\boldsymbol{m}}$ is the data and mask after KNN Imputation, while $\bar{\boldsymbol{x}}'', \bar{\boldsymbol{m}}'$ is the augmented version of $\bar{\boldsymbol{x}}, \bar{\boldsymbol{m}}$, and $k$ is the hyperparameter $k$ in KNN. Next, these inputs are fed into the Encoder network to obtain the augmented and un-augmented embeddings, denoted as $\boldsymbol{z}'$ and $\boldsymbol{z}$ respectively,

$$\boldsymbol{z} = f(\bar{\boldsymbol{x}}, \bar{\boldsymbol{m}}; \boldsymbol{\theta}_E), \ \boldsymbol{z}' = f(\bar{\boldsymbol{x}}'', \bar{\boldsymbol{m}}'; \boldsymbol{\theta}_E); \ \boldsymbol{z}, \boldsymbol{z}' \in \mathbb{R}^{d_e} \tag{2}$$

where $\theta_E$ represents the parameters of the Encoder and $d_e$ is the dimension of the embedding. Following that, the Decoder maps the augmented embedding back to the data space in order to reconstruct the data sample,

$$\hat{\boldsymbol{x}}' = g(\boldsymbol{z}'; \boldsymbol{\theta}_D), \ \hat{\boldsymbol{x}}'^{(v)} \in \mathbb{R}^{d_v} \tag{3}$$

where similarly $\theta_D$ represents the parameters of the Decoder. Simultaneously, clustering is performed using the un-augmented embedding,

$$\boldsymbol{c} = h(\boldsymbol{z}; \boldsymbol{\theta}_C), \ \boldsymbol{c} \in [0, 1]^{d_c} \tag{4}$$

where $\boldsymbol{c}$ is the clustering result, and represents the probabilities of the data sample belonging to $d_c$ cluster centers. During training, the loss function defined in equation 5 is computed to optimize parameters $\boldsymbol{\theta}_E, \boldsymbol{\theta}_D, \boldsymbol{\theta}_C$; During testing, $\boldsymbol{c}$ is regarded as the final clustering result.

In the following sections, we will introduce the Encoder module, including NDE and VDE. The detailed design of the Decoder and the Clustering Module are referenced in **Appendix B.3 and B.4**.

### 3.1.1 NEIGHBOR DIMENSIONAL ENCODER

KNN Imputation 3.2 provides additional information for missing views, but the retrieved nearest neighbors may contain noise and be unreliable. To address this issue, we propose the Neighbor Dimensional Encoder (NDE), which is a series of customized Transformer Encoders (Vaswani et al., 2017), with each one dedicated to a view to fuse its KNN input and filter out noise. The customization of the NDE module mainly involves three key aspects:

**Distance-based Positional Encoding (DPE).** The order or distance of the KNN instances contains vital information regarding the reliability of the inputs, with farther neighbors noisier and less reliable. To capture this information for the permutation invariant Transformer structure, we introduce Distance Positional Encoding (DPE) to provide this extra KNN order information. We explored various designs of positional encoding (PE) considering the data sources and their combination with data in Table 11. Among these configurations, concatenating cosine distance-based (inspired by Nguyen et al. (2021)) or learnable PE with the input yielded the best results. For better interpretability, we chose cosine distance-based as our final design.

**Distance-based Adaptive Masking.** In addition, we propose Distance-based Adaptive Masking (DAM) based on the Transformer's masking mechanism to emphasize the reliability of the input KNN instances, which have a more direct influence than DPE. Originally, the Transformer's mask is binary, with 0 for identity and $-\infty$ for masked. We extend this masking mechanism by introducing a continuous mask with a mapping from the cosine distance. The mapping is an exponential function that maps 0–2 to 0––$\infty$. This masking reduces the weight of distant neighbors in the self-attention mechanism, effectively suppressing noise in the network.

**Output choice.** (Figure 5a) For fusing input sequence information, in typical natural language processing tasks, an additional token like [CLS] is often added (Devlin et al., 2019). However, in our unsupervised task, adding such a meaningless token can introduce noise and lead to performance degradation. Instead, we choose the first output of the NDE module as the result. This design makes the output more correlated with the first input, which is always the most reliable sample in our KNN case. Consequently, important information is emphasized.

For additional formulations and discussions about the NDE, please refer to **Appendix B.1**.

### 3.1.2 VIEW DIMENSIONAL ENCODER

The View Dimensional Encoder (VDE) is designed to fuse view representations and obtain unified embedding. It consists of two parts, with firstly a Feed-Forward Network (FFN) to map the representations of different dimensions to the same latent space, and then followed by a Transformer Encoder for fusion. The FFN consists of three fully connected layers, without normalization or dropout layers, which can be detrimental to the stability of training. The Transformer Encoder is also customized to serve the purpose of view fusion. A key difference between view fusion and KNN fusion is that views are permutation invariant, meaning that changing the order of views should yield the same output. Guided by this principle, we introduce three key customizations:

**Positional Encoding.** Positional Encoding is not used in VDE for the permutation invariant principle.

**Three-level Adaptive Masking.** Similar to the design in NDE, we employ a masking mechanism in VDE to emphasize the reliability of the inputs. As the input is view representations, it can be roughly divided into three categories based on data completeness: (1) complete, (2) missing view with KNN imputation, and (3) missing view without imputation. We design a Three-level Adaptive Mask (TAM) for these categories, the mask values range from completely unmasked (1) to fully masked (3), with an intermediate masking level (2) in between.

**Output choice.** (Figure 5b) To ensure permutation invariance and avoid bias towards any specific views, the embedding is generated by averaging all Transformer output vectors.

For formulations about the View Dimensional Encoder (VDE), please refer to our **Appendix B.2**.

### 3.2 DATA AUGMENTATION AND IMPUTATION

The KNN Imputation algorithm is described in **Appendix B.5**, and below we introduce the data augmentation strategy in our framework.

Our framework is inspired by the denoising auto-encoder, which helps to learn robust representations by introducing noise during training. Three types of noise are designed, including Gaussian noise, random dropout, and view dropout. Gaussian noise helps prevent overfitting by introducing variability in the input data. Random dropout, functional as a regularization technique, encourages the model to learn more robust features by forcing it to rely on different subsets of the input data. View dropout is a noise specifically designed for the IMVC task. It randomly drops out (or masks) one or more views from the input data during each training iteration. This encourages the model to learn representations that are more robust to missing views and helps improve the performance of the model on incomplete multi-view datasets. The formulation is placed in the **Appendix B.6**.

### 3.3 TRAINING STRATEGY AND LOSS FUNCTION

#### 3.3.1 LOSS FUNCTION

In the training process, we utilize a combination of three loss functions formulated as follows:

$$L(\boldsymbol{x}, \hat{\boldsymbol{x}}', \boldsymbol{z}, \boldsymbol{z}', \boldsymbol{c}) = L_{rec}(\boldsymbol{x}, \hat{\boldsymbol{x}}') + \lambda_1 L_{aug}(\boldsymbol{z}, \boldsymbol{z}') + \lambda_2 L_{clu}(\boldsymbol{c}) \tag{5}$$

$L_{rec}$ (equation 21): This loss term corresponds to the reconstruction loss of the auto-encoder. By minimizing this loss, the auto-encoder learns to reconstruct the input data accurately. $L_{aug}$ (equation 22): The embedding robustness loss, encourages the learned representations to be consistent when augmentations are applied, promoting the robustness of the learned representations. $L_{clu}$ (equation 23): The DEC-based clustering loss. This loss optimizes embeddings for clustering with gradients from high-confidence samples. Hyperparameters $\lambda_1$ and $\lambda_2$ control the balance between different loss components. By jointly minimizing the three loss terms, our network can learn representations that are both informative and clustering-friendly. For the formulation of each loss function, please refer to our **Appendix B.7**.

#### 3.3.2 TRAINING STRATEGY

The training process is divided into two stages. In the first stage, the auto-encoder is pre-trained using $L_{rec}$ and $L_{aug}$, focusing on learning robust representations. Once the pre-training is complete, the Clustering Module is initialized. In the second stage, $L_{clu}$ is added for joint training to learn a clustering-friendly representation. Experiments show that the DEC-based method is sensitive to the initialization of the cluster centers. To address this issue, we proposed a Dynamic Initialization strategy. An unsupervised metric score $s$ based on the Davies-Bouldin Index (DBI) and the standard deviation of cluster sizes is introduced to evaluate the quality of checkpoints. The checkpoint with the best score is eventually used for initializing the cluster centers. For a detailed description of the training process, including the formulation of the unsupervised metric score $s$ and the training Algorithm 2, please refer to the corresponding sections in our **Appendix B.8**.

## 4 EXPERIMENTS

Please refer to **Appendix C.1** for the hyperparameter settings and design details in our experiments.

### 4.1 DATASETS AND METRICS

Experiments were performed on four multi-view datasets to validate the effectiveness of our method. The dataset characteristics are summarized in Table 1. We report the widely used metrics Clustering Accuracy (Acc), Normalized Mutual Information (NMI), and Adjusted Rand Index(ARI) as results. We run each experiment 10 times and report the average value and standard deviation (after $\pm$). Details about our experiment and view-missing settings can be found in **Appendix C.2**.

Table 1: The statistic of 4 datasets used in our experiments.

| Name | Views | Clusters | Samples | Dimensions |
|---|---|---|---|---|
| Handwritten (Duin, 2023) | 6 | 10 | 2000 | 240/76/216/47/64/6 |
| Caltech101-7 (Cai et al., 2013) | 5 | 7 | 1400 | 40/254/1984/512/928 |
| ALOI_Deep (Liu et al., 2023) | 3 | 100 | 10800 | 2048/4096/2048 |
| Scene15 (Fei-Fei & Perona, 2005; Cai et al., 2013) | 2 | 15 | 4485 | 20/59 |

Table 2: Comparison of our method with state-of-the-art approaches on 4 benchmark datasets. A fixed missing rate $m_r = 0.5$ is applied to all datasets. The best result is highlighted in **bold** while the suboptimal is underlined.

| Datasets | Handwritten | | | Caltech101-7 | | |
|---|---|---|---|---|---|---|
| Metrics | Acc(%) | NMI(%) | ARI(%) | Acc(%) | NMI(%) | ARI(%) |
| Completer (Lin et al., 2021) | $55.67 \pm 4.78$ | $54.73 \pm 3.08$ | $34.11 \pm 4.12$ | $56.21 \pm 8.64$ | $52.07 \pm 5.51$ | $33.72 \pm 8.20$ |
| DSIMVC (Tang & Liu, 2022) | $76.65 \pm 5.91$ | $71.33 \pm 3.42$ | $63.67 \pm 5.59$ | $67.76 \pm 3.76$ | $57.12 \pm 1.99$ | $49.10 \pm 2.92$ |
| SURE (Yang et al., 2023b) | $65.47 \pm 6.27$ | $61.85 \pm 4.98$ | $50.41 \pm 7.14$ | $68.14 \pm 5.50$ | $53.63 \pm 5.26$ | $47.69 \pm 6.81$ |
| DCP (Lin et al., 2023) | $68.10 \pm 4.51$ | $68.88 \pm 2.16$ | $50.94 \pm 6.25$ | $61.68 \pm 8.21$ | $62.99 \pm 5.33$ | $45.71 \pm 9.44$ |
| CPSPAN (Jin et al., 2023) | $89.27 \pm 3.78$ | $83.18 \pm 2.01$ | $\underline{79.98 \pm 3.70}$ | $\underline{77.62 \pm 4.74}$ | $\underline{69.70 \pm 4.04}$ | $\underline{63.23 \pm 5.51}$ |
| RecFormer (Liu et al., 2023) | $\underline{89.59 \pm 0.77}$ | $\underline{80.30 \pm 1.16}$ | $78.02 \pm 1.55$ | $73.39 \pm 2.45$ | $63.02 \pm 2.40$ | $57.54 \pm 2.97$ |
| URRL-IMVC (ours) | $\mathbf{94.66 \pm 0.33}$ | $\mathbf{88.68 \pm 0.52}$ | $\mathbf{88.57 \pm 0.68}$ | $\mathbf{92.95 \pm 2.60}$ | $\mathbf{86.29 \pm 1.76}$ | $\mathbf{86.02 \pm 2.91}$ |

| Datasets | ALOI_Deep | | | Scene15 | | |
|---|---|---|---|---|---|---|
| Metrics | Acc(%) | NMI(%) | ARI(%) | Acc(%) | NMI(%) | ARI(%) |
| Completer (Lin et al., 2021) | $43.56 \pm 3.42$ | $72.92 \pm 1.65$ | $19.39 \pm 2.11$ | $39.09 \pm 2.09$ | $\mathbf{42.04 \pm 1.79}$ | $\underline{23.35 \pm 1.72}$ |
| DSIMVC (Tang & Liu, 2022) | $71.98 \pm 2.25$ | $90.53 \pm 0.65$ | $69.09 \pm 1.78$ | $29.75 \pm 1.51$ | $32.97 \pm 0.90$ | $15.62 \pm 0.68$ |
| SURE (Yang et al., 2023b) | $50.77 \pm 4.70$ | $86.51 \pm 1.45$ | $42.07 \pm 6.20$ | $38.83 \pm 2.25$ | $37.24 \pm 0.55$ | $20.98 \pm 1.09$ |
| DCP (Lin et al., 2023) | $58.45 \pm 7.41$ | $87.60 \pm 2.77$ | $53.05 \pm 12.27$ | $38.27 \pm 1.66$ | $41.46 \pm 0.93$ | $22.39 \pm 1.27$ |
| CPSPAN (Jin et al., 2023) | $74.08 \pm 3.91$ | $92.54 \pm 1.13$ | $72.86 \pm 3.04$ | $37.46 \pm 1.91$ | $\underline{41.61 \pm 1.86}$ | $22.80 \pm 1.65$ |
| RecFormer (Liu et al., 2023) | $\underline{85.45 \pm 1.79}$ | $\underline{96.52 \pm 0.24}$ | $\underline{84.85 \pm 1.14}$ | $33.90 \pm 1.45$ | $34.34 \pm 1.18$ | $17.10 \pm 1.12$ |
| URRL-IMVC (ours) | $\mathbf{92.33 \pm 0.70}$ | $\mathbf{97.93 \pm 0.22}$ | $\mathbf{91.98 \pm 0.71}$ | $\mathbf{41.69 \pm 1.60}$ | $40.88 \pm 0.92$ | $\mathbf{23.93 \pm 1.11}$ |

### 4.2 COMPARISON WITH STATE-OF-THE-ARTS

We compare our approach with several state-of-the-art approaches listed in Table 2. Other comparisons about textual datasets, different view numbers, traditional IMVC methods, model parameters, and computational costs can be found in **Appendix C.3**.

**Comparison on different datasets.** URRL-IMVC achieved state-of-the-art performance on the four benchmark datasets, surpassing most existing approaches, as indicated in Table 2. Our approach consistently outperformed other SOTA methods across all evaluation metrics, except for the NMI on the Scene15 dataset (discussion in **Appendix C.4**). Additionally, URRL-IMVC exhibited stability compared to other SOTA methods, with a relatively low standard deviation across 10 experiments. This excellent clustering performance and stability can be attributed to our unified embedding approach, which effectively captures the underlying data structure while remaining robust in the presence of missing views. Notably, our approach excelled on datasets with a larger number of views, such as Handwritten and Caltech101-7, thanks to its ability to leverage the consensus and complementary information in multi-view data. This advantage mitigates the limitations of cross-view contrastive learning as the number of views increases.

**Comparison with different missing rates.** As depicted in Figure 3, URRL-IMVC consistently outperformed other approaches, establishing an upper bound for clustering performance regardless of the missing rate ($m_r$). Our approach displayed better stability compared to other methods, with a gradual decrease in accuracy as the missing rate increased. In contrast, other approaches exhibited

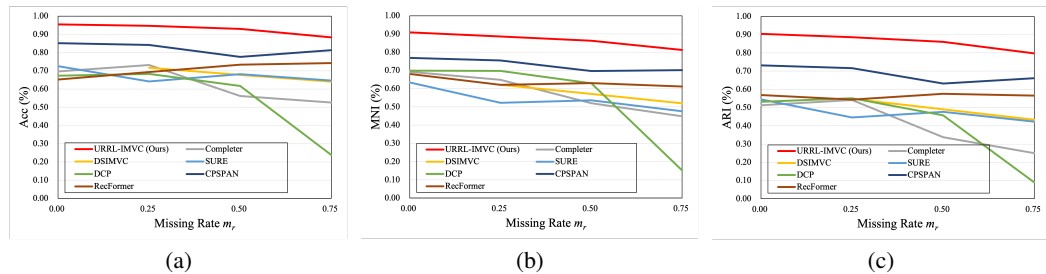

Figure 3: Comparison with state-of-the-art approaches under different missing conditions on the Caltech101-7 dataset. The performance of each approach is reported using fold lines.

more fluctuation, rendering their results less predictable. Notably, DCP and Completer experienced a significant decline in performance when the missing rate reached 0.75, as they only trained their cross-view contrastive and recovery networks using complete samples. Insufficient training samples led to unsatisfactory recovery outcomes and fragile representations for clustering. In contrast, our approach focused on the robustness of the unified representation, allowing us to circumvent these limitations and achieve stable and high performance across varying missing rates.

## 4.3 Ablation Studies

Unless otherwise specified, the experiments were conducted on the Caltech101-7 dataset with the missing rate $m_r = 0.5$. In certain experiments, the Clustering Module and Dynamic Initialization were disabled to provide clearer observations of specific phenomena. Additional ablation studies regarding detailed designs (e.g., cluster initialization) and hyperparameters (e.g., $k$ in KNN Imputation) can be found in **Appendix C.5 and C.6**.

### 4.3.1 Ablation on Modules

Table 3: Ablation study on our designed modules. We begin with a baseline model, which consists of a simple Transformer-based auto-encoder with binary masks. We then gradually incorporate our designed modules and report the metrics along with their increments. "KNN": KNN Imputation; "Aug": data augmentation and robustness loss; "DPE&AM": DPE, DAM and TAM described in NDE and VDE; "CM": clustering module and clustering loss; "DI": Dynamic Initialization for CM.

| Dataset | Baseline | + KNN | + Aug | + DPE&AM | + CM | + DI |
|---|---|---|---|---|---|---|
| Acc | $76.93 \pm 4.35$ | $83.68 \pm 2.79$ | $84.77 \pm 3.59$ | $85.25 \pm 4.82$ | $91.73 \pm 4.23$ | $92.95 \pm 2.60$ |
| Increment | | +6.75 | +1.09 | +0.48 | +6.48 | +1.22 |
| NMI | $64.79 \pm 2.18$ | $72.76 \pm 2.21$ | $74.92 \pm 2.54$ | $75.91 \pm 3.33$ | $85.33 \pm 3.18$ | $86.29 \pm 1.76$ |
| Increment | | +7.97 | +2.15 | +0.99 | +9.42 | +0.96 |
| ARI | $56.52 \pm 3.99$ | $68.60 \pm 4.48$ | $71.76 \pm 3.69$ | $72.61 \pm 5.50$ | $84.70 \pm 4.62$ | $86.02 \pm 2.91$ |
| Increment | | +12.08 | +3.16 | +0.85 | +12.09 | +1.33 |

In Table 3, we present the results of our ablation study on the main modules we designed. The results show that KNN Imputation had significant improvements compared to the baseline, and augmentation further boosted the performance. Note that KNN Imputation and augmentation actually had approximately equal contributions, the difference in increment comes from the order they are applied. Additionally, KNN Imputation seemed to enhance the stability of the performance, while augmentation had the opposite effect. This aligns with our expectations, as the former provides additional information, leading to a more stable training process, while the latter adds constraint information, resulting in a more unstable training process. The incorporation of Distance-based Positional Encoding and Adaptive Masking slightly improved the performance. Although the improvements were modest, these modules contributed to capturing the underlying data structure more effectively. The introduction of the Clustering Module significantly improves the clustering performance. However, it did not improve the stability. This is consistent with our observation that DEC-based training

is sensitive to initialization and cannot achieve good clusters with poor initialization. Finally, the inclusion of Dynamic Initialization partially addressed this issue and improved both stability and accuracy. Overall, the ablation study provides valuable insights into the contributions and effects of each module on the performance of our approach.

### 4.3.2 VISUALIZATION

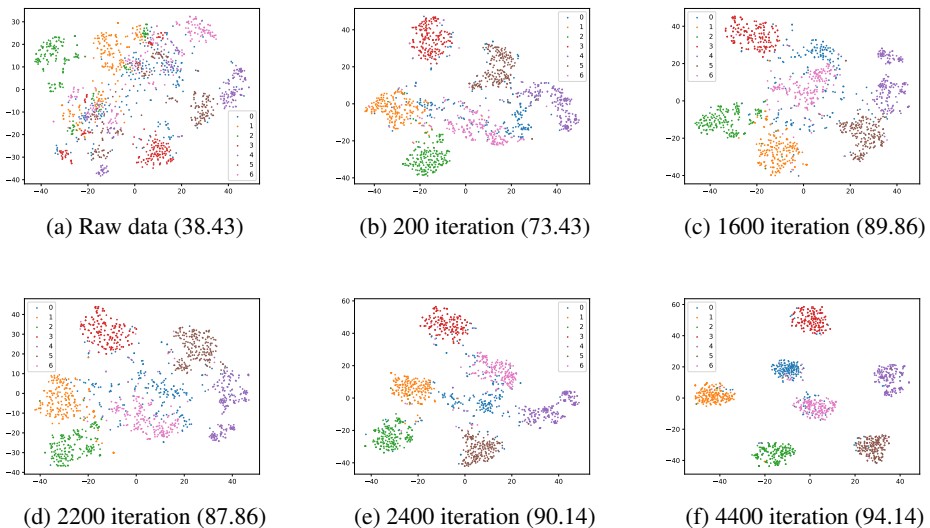

(a) Raw data (38.43)       (b) 200 iteration (73.43)       (c) 1600 iteration (89.86)

(d) 2200 iteration (87.86)       (e) 2400 iteration (90.14)       (f) 4400 iteration (94.14)

Figure 4: T-SNE visualization of the embeddings during the training process on the Caltech101-7 dataset. The iteration number and corresponding accuracy, is recorded below each sub-figure. The training process consists of 4400 iterations, with the Clustering Module initialized at 2200 iterations.

Figure 4 presents a T-SNE visualization of the embeddings during one training process. Initially, in Figure 4a, the visualization of the multi-view raw data concatenated as embeddings appears to be disorganized. After 200 iterations of training, in Figure 4b, inherent structures start to be captured, and the local accuracy peak value (89.86) occurs at 1600 iterations 4c. At 2200 iterations, the Clustering Module is initialized, and joint training with DEC-based clustering loss commences. Note that according to our dynamic initialization strategy, the checkpoint with the best unsupervised metric score 24, is loaded for joint training, which is 1600 (better than 2200) in this case. After 200 iterations of joint training, as depicted in Figure 4e, the clusters become more compact. Finally, at the end of the training, as shown in Figure 4f, the clusters become very compact, numerous samples initially misclustered with low confidence are now corrected, and the accuracy reaches 94.14. For visualization of loss and clustering performance throughout training, please refer to **Appendix C.7**.

## 5 CONCLUSION

In this paper, we proposed URRL-IMVC, a novel unified and robust representation learning framework for the incomplete multi-view clustering task. By leveraging carefully designed modules, including neighbor dimensional encoders and view dimensional encoders, we successfully fuse the multi-view information into a unified embedding. URRL-IMVC offers a more comprehensive solution compared to potentially limiting cross-view contrastive learning. Through the utilization of KNN imputation and data augmentation strategies, we directly acquire robust embeddings that effectively handle the view-missing condition, eliminating the need for explicit missing view recovery and its associated computation and unreliability. Furthermore, incremental improvements, such as adaptive masking and dynamic initialization, significantly enhance the clustering stability and performance, achieving state-of-the-art results. This improved robust and unified representation learning framework acts as a powerful tool for addressing the challenges of IMVC and provides valuable insights for future research in this domain.

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

# A    RELATED WORKS APPENDIX

## A.1    TRADITIONAL IMVC APPROACHES

In the past decade, extensive research has been conducted on the Incomplete Multi-view Clustering (IMVC) task, leading to the proposal of various solutions (Wen et al., 2023), mainly including matrix factorization-based, kernel-based and graph-based methods. Matrix factorization-based approaches, such as Partial Multi-view Clustering (PMVC) (Li et al., 2014), were among the earliest studies in this field. Zhao et al. (2016) and Xu et al. (2018) extended the idea of matrix factorization by incorporating graph embedding techniques. To handle more complex missing conditions, weighted matrix factorization-based approaches have been proposed, including (Shao et al., 2015; 2016; Hu & Chen, 2018; 2019), which demonstrate improved performance in handling complex missing data scenarios. Kernel-based IMVC approaches focus on learning consensus representations or multiple representations from multiple kernels, which capture different aspects of the data. For instance, Trivedi et al. (2010) proposed Laplacian regularization and kernel canonical correlation analysis for solving the IMVC problem. Multiple kernel K-means-based approaches, such as (Liu et al., 2017; Ye et al., 2017; Zhu et al., 2018; Liu et al., 2020), have also been proposed. Graph-based approaches aim to obtain a consensus graph or representation by combining information from multiple incomplete graphs from each view, facilitating a better understanding of the underlying data structure. Gao et al. (2016) proposed recovering missing latent representations via co-training to obtain the consensus representation. Wang et al. (2019a) introduced spectral perturbation theory to address the problem, providing insights into the impact of perturbations on graph Laplacians. Additionally, Wen et al. (2020) presented adaptive graph construction for learning consensus representation, which dynamically adjusts the graph structure based on the data. CGMAA (Li et al., 2023b) generated bipartite graphs for clustering and proposed to address the cross-view anchor misalignment problem by predefining an anchor graph according to the prior anchor information. Yang et al. (2023a) proposed CGMIMC to solve the IMVC task by constructing connection graphs and adopting cross-view graph matching. PIMVC proposed by Deng et al. (2023) formulated a graph-regularized projective consensus representation learning model with graph constraint to overcome the drawbacks of matrix factorization-based methods. We also noted that some other methods with different thinking have also emerged. For example, Multi-view Probabilistic Clustering (MPC) (Liu et al., 2022) proposes a multi-view fusion method based on posterior matching probabilities, which achieves very competitive results on the IMVC task. These aforementioned approaches represent significant contributions to the IMVC field and have paved the way for further advancements. These aforementioned approaches represent significant contributions to the IMVC field and have paved the way for further advancements.

# B    METHOD APPENDIX

## B.1    NEIGHBOR DIMENSIONAL ENCODER - FORMULATION AND DISCUSSION

The Neighbor Dimensional Encoder (NDE) can be formulated as,

$$\boldsymbol{x}_N = \{\boldsymbol{x}_N^{(1)}, \boldsymbol{x}_N^{(2)}, \cdots, \boldsymbol{x}_N^{(V)}\},\ \boldsymbol{x}_N^{(v)} \in \mathbb{R}^{d_v} \tag{6}$$

$$\boldsymbol{x}_N^{(v)} = f_N^{(v)}(DPE(\bar{\boldsymbol{x}}^{(v)}), DAM(\bar{\boldsymbol{x}}^{(v)}); \boldsymbol{\theta}_N^{(v)})_{[0,:]} \tag{7}$$

It can be observed from Fig 2 and the above equations that a separate network is used for each view as the input dimension varies. The basic structure of NDE follows the structure of the Transformer Encoder (Vaswani et al., 2017). The $v$th Transformer Encoder corresponding to the $v$th view is represented with $f_N^{(v)}(\cdot; \boldsymbol{\theta}_N^{(v)})$ in equation 7, and $\boldsymbol{\theta}_N^{(v)}$ is its parameters. It takes two inputs $DPE(\bar{\boldsymbol{x}}^{(v)})$ and $DAM(\bar{\boldsymbol{x}}^{(v)})$, they are the input sequence and the mask respectively, generated

with Distance-based Positional Encoding (DPE) and Distance-based Adaptive Masking (DAM) operation described below. As described in section 3.1.1, we only take the first vector from the output sequence, which is denoted by $[0, :]$.

The DPE can be explained as,

$$DPE(\bar{\boldsymbol{x}}^{(v)}) = \bar{\boldsymbol{x}}^{(v)} \oplus d(\bar{\boldsymbol{x}}^{(v)}), \ d(\bar{\boldsymbol{x}}^{(v)}) \in \mathbb{R}^{k \times k} \tag{8}$$

in which $d()$ is the function calculating the pair-wise distance of $k$ vectors and return a $k \times k$ distance matrix, and $\oplus$ stands for matrix concatenation. For the pair-wise distance function, we tried cosine distance and Euclidean distance and found cosine distance more suitable. Given two input vectors $\boldsymbol{x}_1$ and $\boldsymbol{x}_2$, the cosine distance is formulated as,

$$d_{cos}(\boldsymbol{x}_1, \boldsymbol{x}_2) = 1 - \frac{\boldsymbol{x}_1 \cdot \boldsymbol{x}_2}{||\boldsymbol{x}_1|| \cdot ||\boldsymbol{x}_2||} \tag{9}$$

We conjecture that cosine distance is more suitable for two reasons. First, its value range is 0–2, which is more stable than Euclidean distance. Second, as we are using zero vectors to represent missing samples, the distance between missing samples and any other samples is fixed to 1 with cosine distance, which also helps stabilize the training.

The masked self-attention in Transformer can be described as,

$$Attention(Q, K, V, M_N) = softmax(\frac{QK^T}{\sqrt{d_k}} + M_N)V \tag{10}$$

where $Q, K, V$ stands for Query, Key, and Value matrix respectively, and $M_N$ is the mask applied.

In our situation, we are looking for a method to reduce the influence of noisy neighbors, while emphasizing high-confidence neighbors, so Distance-based Adaptive Masking is proposed. Like in the DPE, we map the cosine distance to the mask used in this Transformer Encoder. For mapping 0–2 to 0–(-inf), the mapping function is designed based on the exponential function,

$$M_N = DAM(\bar{\boldsymbol{x}}^{(v)}) = -e^{\alpha \cdot d(\bar{\boldsymbol{x}}^{(v)})_{[0,:]} + \beta} \tag{11}$$

where $d(\bar{\boldsymbol{x}}^{(v)})_{[0,:]}$ is the first row of the distance matrix, representing the cosine distance between each neighbor and the center sample, $\alpha$ and $\beta$ are two hyperparameters that control the mapping range. After the mapping, the mask is broadcasted and applied to the neighbor sample to lower its weight in self-attention adaptively.

## B.2 VIEW DIMENSIONAL ENCODER - FORMULATION

As depicted in Figure 2, the VDE consists of a Feed Forward Network (FFN) part and a Transformer Encoder part. The FFN part of VDE for mapping embeddings to the same latent space can be described as,

$$\boldsymbol{x}_A = \boldsymbol{x}_A^{(1)} \oplus \boldsymbol{x}_A^{(2)} \oplus \cdots \oplus \boldsymbol{x}_A^{(N)}, \ \boldsymbol{x}_A \in \mathbb{R}^{V \times d_e} \tag{12}$$

$$\boldsymbol{x}_A^{(v)} = \sigma(\sigma(\boldsymbol{x}_N^{(v)} \boldsymbol{W}_1^{(v)} + \boldsymbol{b}_1^{(v)}) \boldsymbol{W}_2^{(v)} + \boldsymbol{b}_2^{(v)}) \boldsymbol{W}_3^{(v)} + \boldsymbol{b}_3^{(v)} \tag{13}$$

In the equation, $\oplus$ represents the concatenate operation, $\sigma$ is the activation function, $\boldsymbol{W}$ and $\boldsymbol{b}$ are the weight matrix and bias vector of the fully connected (FC) layer respectively.

The Transformer Encoder part of the VDE can be explained as,

$$\boldsymbol{z} = \sum_{v=1}^{V} f_V(\boldsymbol{x}_A, \bar{\boldsymbol{m}}, \boldsymbol{m}; \boldsymbol{\theta}_V)/V \tag{14}$$

in which $f_V(\cdot; \boldsymbol{\theta}_A)$ is the VDE Transformer Encoder structure. As mentioned in section 3.1.2, the output is averaged as the unified embedding $\boldsymbol{z}$.

The Three-level Adaptive Masking (TAM) used in the Transformer Encoder part is formulated as,

$$M_V = \begin{cases} 0, \ \boldsymbol{m}_j = 1 \\ \gamma, \ \sum_{i=1}^{k} \bar{\boldsymbol{m}}_{ij} > 0 \ \& \ \boldsymbol{m}_j = 0 \\ -\infty, \ \sum_{i=1}^{k} \bar{\boldsymbol{m}}_{ij} = 0 \ \& \ \boldsymbol{m}_j = 0 \end{cases} \tag{15}$$

in which $\bar{\boldsymbol{m}}$ and $\boldsymbol{m}$ are the KNN imputation generated and the original missing matrix. $\gamma$ is a negative hyperparameter to control the emphasizing intensity.

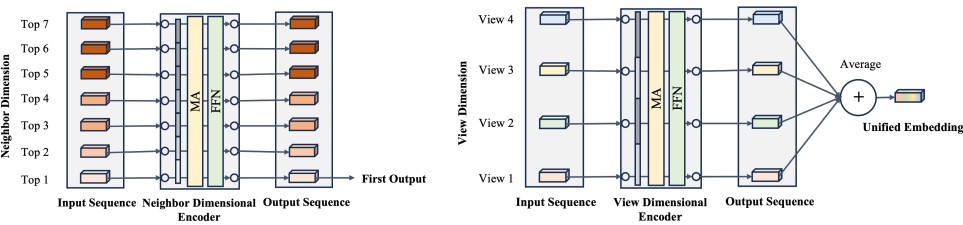

(a) Neighbor Dimensional Encoder        (b) View Dimensional Encoder

Figure 5: An intuitive visualization of the output choice of the Neighbor Dimensional Encoder (NDE) and View Dimensional Encoder (VDE). For VDE, we only illustrated the Transformer Encoder part. In NDE, the first of the output sequence is chosen to provide a bias on the most reliable input. In VDE, the outputs are fused to provide an unbiased representation of all views.

### B.3 DECODER

The Decoder in our model is designed as a compact 4-layer FFN, to reconstruct the input. Similar to the FFN in VDE, we removed its normalization and dropout layer for better stability. Through our experiments, we have observed that a deep and complex Decoder does not necessarily improve the clustering performance and may even have negative effects in certain cases. One possible explanation for this phenomenon is that a shallow and simple Decoder serves as a regularization technique on the embedding space. By establishing a simple mapping from the embedding space to the data space, the shallow Decoder prevents the embedding space from collapsing. This regularization effect is similar to the Locality-preserving Constraint proposed by Huang et al. (2014), which helps preserve the local structure of the data. The process of Decoder is formulated as,

$$\hat{\boldsymbol{x}} = \{\hat{\boldsymbol{x}}^{(1)}, \hat{\boldsymbol{x}}^{(2)}, \cdots, \hat{\boldsymbol{x}}^{(V)}\}, \ \hat{\boldsymbol{x}}^{(v)} \in \mathbb{R}^{d_v} \tag{16}$$

$$\hat{\boldsymbol{x}}^{(v)} = g^{(v)}(\boldsymbol{z}; \theta_D^{(v)}) \tag{17}$$

### B.4 CLUSTERING MODULE - DESCRIPTION AND FORMULATION

The auto-encoder we have designed allows us to obtain robust representations and capture the inherent structures of data. However, these inherent structures may not necessarily follow a cluster-oriented distribution. To enhance the clustering performance, we introduce a Clustering Module inspired by DEC (Xie et al., 2016). Additionally, we design a dynamic initialization strategy, as described in Appendix B.8, to improve the clustering stability by finding better initialization. Below we describe the procedure of the DEC-based Clustering Module. First, a traditional clustering method is used to generate the cluster centers $\boldsymbol{\theta}_C = \boldsymbol{\mu} \in \mathbb{R}^{d_c \times d_e}$ from embedding $\boldsymbol{z}$. Then, the similarity matrix $\boldsymbol{q} \in [0,1]^{N \times d_c}$ is generated with Student's t-distribution, in which $q_{ij}$ represents the possibility sample $i$ belongs to cluster $j$, and $\boldsymbol{c} \in \mathbb{R}^{d_c}$ defined in equation 4 is a row of $\boldsymbol{q}$. With the similarity matrix $\boldsymbol{q}$, we calculate the target distribution $\boldsymbol{p}$ as,

$$p_{ij} = \frac{q_{ij}^2 / f_j}{\sum_{j=1}^{d_c} (q_{ij}^2 / f_j)} \tag{18}$$

in which $f_j$ is the soft cluster size $\sum_{i=0}^{N-1} q_{ij}$. The training target is the KL-Divergence between $\boldsymbol{p}$ and $\boldsymbol{q}$ (equation 23) which is defined in Appendix B.7. Our improvement on the initialization of the Clustering Module is explained in Appendix B.8. Finally, the clustering result can be obtained by finding the maximum possibility in each row of $\boldsymbol{q}$.

### B.5 KNN IMPUTATION - FORMULATION

Below we introduce the KNN Imputation operation described in equation 1 and 19. The KNN search is conducted separately in each view, taking into account the incomplete condition. For

existing views, the KNN is directly obtained, while for missing views, existing views from the same sample are used for search KNN. Specifically, for a missing view $\boldsymbol{x}_i^{(v)}$ of sample $\boldsymbol{x}_i$, we first find all existing views $\boldsymbol{x}_i^{(b)}$ of the same sample. Then we iterate through the KNN samples $\boldsymbol{x}_j^{(b)}$ of these existing views to check if $\boldsymbol{x}_j^{(v)}$ exists. If it does, we appended it to the KNN list of the missing view. Finally, we select the top $k$ samples from the KNN list as the imputation for the missing view. If the length of the KNN list is less than $k$, then the remaining positions are filled with zeros. The detailed procedures are listed in Algorithm 1. Along with the KNN imputation $\bar{\boldsymbol{x}}$, a missing indicator matrix $\bar{\boldsymbol{m}} \in \mathbb{R}^{k \times V}$ is generated, where 1 represents a position filled with a KNN sample and 0 represents a position filled with zeros.

---

**Algorithm 1** Procedure of KNN Imputation

---

**Input:** Target $\boldsymbol{x}_i^{(v)}$, which is the $v$th view of the $i$th sample, dataset $X$, missing indicator matrix $M$, hyperparameter $k$.
1: **if** $M_{iv} = 1$ **then**                                      ▷ The view exists
2:      Return $\boldsymbol{x}_i^{(v)}$'s KNN
3: **else**                                            ▷ The view is missing
4:      $a = 0$, create empty KNN list
5:      **while** $a < k$ **do**                               ▷ Traverse $k$ neighbors
6:          $b = 1$
7:          **while** $b <= V$ **do**                       ▷ Traverse all views
8:              **if** $b = v$ or $M_{ib} = 0$ **then**      ▷ $b$th view of target sample $\boldsymbol{x}_i$ is missing
9:                  pass
10:             **else**                      ▷ $b$th view of target sample $\boldsymbol{x}_i$ exists
11:                  Find $a$th nearest neighbor of $\boldsymbol{x}_i^{(b)}$, denoted as $\boldsymbol{x}_j^{(b)}$
12:                  **if** $M_{jv} = 0$ **then**           ▷ $v$th view of neighbor $\boldsymbol{x}_j$ is missing
13:                      pass
14:                  **else**                   ▷ $v$th view of neighbor $\boldsymbol{x}_j$ exists
15:                      Add $\boldsymbol{x}_j^{(v)}$ to KNN list
16:                  **end if**
17:             **end if**
18:             $b = b + 1$
19:          **end while**
20:          $a = a + 1$
21:      **end while**
22:      **if** KNN list length $< k$ **then**
23:          Pad KNN list with zeros to length $k$
24:      **else**
25:          Choose top $k$ from KNN list
26:      **end if**
27:      Return KNN list
28: **end if**
     **Output:** KNN Imputation $\bar{\boldsymbol{x}}_i^{(v)}$

---

### B.6   DATA AUGMENTATION - FORMULATION

Below are the equations for the three kinds of augmentation we used. By combining equation 19 and 20, the KIDA operation in equation 1 is obtained.

$$\bar{\boldsymbol{x}}', \bar{\boldsymbol{m}}' = IKNN(\boldsymbol{x}, \boldsymbol{m} \odot \boldsymbol{m}_{vd}), \ P(\boldsymbol{m}_{vd} = 0) = \phi_1 \tag{19}$$

$$\bar{\boldsymbol{x}}'' = (\bar{\boldsymbol{x}}' + \phi_2 \boldsymbol{n}) \odot \boldsymbol{m}_{rd}, \ \boldsymbol{n} \sim \mathcal{N}(0,1), \ P(\boldsymbol{m}_{rd} = 0) = \phi_3 \tag{20}$$

The view dropout augmentation is shown in equation 19, where random views are masked or dropped out with a possibility of $\phi_1$, before the KNN Imputation (Algorithm 1) step. The dropout mask is applied with element-wise multiplication, denoted by $\odot$. The masked views are regarded as missing views in both the KNN Imputation and the auto-encoder network. After the KNN Imputation, in equation 20, Gaussian noise is added to the input data to introduce variability, whose

intensity is controlled by a hyperparameter $\phi_2$. After that, random values in the input are set to zero with a probability of $\phi_3$, which is the random dropout augmentation. Finally, the augmented input $\bar{x}''$ along with the corresponding missing indicator matrix $\bar{m}'$ can be used for training. By incorporating these noise augmentation techniques, our approach enhances the robustness of the model to missing views, introduces variability in the input data, and helps prevent overfitting during the training process.

### B.7 LOSS - FORMULATION

Below we introduce the formulation of the three loss terms. The first is the reconstruction loss of the auto-encoder, denoted as,

$$L_{rec}(\boldsymbol{x}, \hat{\boldsymbol{x}}') = \sum_{v=1}^{V}(||\hat{\boldsymbol{x}}'^{(v)} - \boldsymbol{x}^{(v)}||^2 \times \boldsymbol{m}_v) \tag{21}$$

The missing indicator $\boldsymbol{m}_v$ is multiplied so only the mean square errors of existing views are calculated. Note that during training, the network output is $\hat{\boldsymbol{x}}'$ as the input is the augmented input $\bar{\boldsymbol{x}}''$ (equation 2 and 3), so the network learns to reconstruct masked views with KNN hints and cross-view correlation and thus, learns to predict missing views' information implicitly. The second loss is the embedding robustness loss against augmentation.

$$L_{aug}(\boldsymbol{z}, \boldsymbol{z}') = -log\frac{e^{-||\boldsymbol{z}'_i-\boldsymbol{z}_i||}}{\sum_{j=1}^{B} e^{-||\boldsymbol{z}'_i-\boldsymbol{z}_j||}} \tag{22}$$

During training, we infer both augmented input $\boldsymbol{x}''$ and original input $\boldsymbol{x}$, and obtain the corresponding embeddings $\boldsymbol{z}'$ and $\boldsymbol{z}$. Our robustness goal is equivalent to minimizing the distance between $\boldsymbol{z}'$ and $\boldsymbol{z}$ (equation 2). As we are using mini-batched training, we designed the above loss term based on cross-entropy loss. By minimizing the loss function, the distance between $\boldsymbol{z}'_i$ and $\boldsymbol{z}_i$ is minimized while the distances between $\boldsymbol{z}'_i$ and embeddings of other samples $\boldsymbol{z}_j, j \neq i$ are maximized. This design prevents the embedding space from collapsing. The third loss is the DEC-based clustering loss (Xie et al., 2016) used in the Clustering Module (Appendix B.4), which is the KL divergence between distribution $\boldsymbol{p}'$ and $\boldsymbol{q}'$ computed from augmented input $\boldsymbol{x}''$ during training.

$$L_{clu}(\boldsymbol{c}) = KL(\boldsymbol{p}'||\boldsymbol{q}') = \sum_{i=1}^{B}\sum_{j=1}^{d_c} p'_{ij}log\frac{p'_{ij}}{q'_{ij}} \tag{23}$$

The final loss is the weighted sum of the above three loss terms, formulated in 3.3

### B.8 TRAINING STRATEGY AND ALGORITHM

Below we introduce the Dynamic Initialization strategy and Algorithm 2 to describe the detailed training procedure.

During auto-encoder pre-training, several checkpoints are saved. Then during Cluster Module initialization (Appendix B.4), an unsupervised metric is introduced to determine which checkpoint shows the best clustering structure and load it for joint training. The unsupervised metric score $s$ is designed as,

$$s = DBI(\boldsymbol{c}, \boldsymbol{z}) + \omega \cdot \varsigma(\sum_{i=1}^{N}\boldsymbol{c}_{ij})/\mu(\sum_{i=1}^{N}\boldsymbol{c}_{ij}) \tag{24}$$

The metric consists of two terms. The first term is the Davies–Bouldin Index $DBI()$ for measuring the clustering performance, and the second term is the normalized standard deviation of cluster size for avoiding trivial solutions. $\sum_{i=1}^{N}\boldsymbol{c}_{ij}$ is the cluster size vector, we calculate its standard deviation $\varsigma$ and normalize with average size $\mu$, $\omega$ is a hyperparameter to control the balance between the two terms. We choose the checkpoint with the lowest $s$ for Clustering Module initialization and joint training.

The training process is controlled by three hyperparameters: $E_p$, which represents the number of pre-training epochs, $E_j$, which represents the number of joint training epochs, and $E_c$, which represents the starting point at which the checkpoints for dynamic initialization are saved.

**Algorithm 2** Training process of URRL-IMVC

**Input:** Dataset $X$, missing indicator matrix $M$, hyperparameters.

1: Initialize model parameters $\theta_E$, $\theta_D$, and $\theta_C$. Pre-compute KNN-search results. epoch = 0, iteration per epoch = $I_e$, checkpoint save interval = $E_s$

2: **while** $epoch < E_p$ **do** $\qquad\qquad\qquad\qquad\qquad\qquad\qquad$ ▷ Stage 1: Pre-training

3: $\quad$ iteration = 0

4: $\quad$ **while** $iteration < I_e$ **do**

5: $\qquad$ **Pre-process**: KNN Imputation and Data Augmentation by equation 19 and 20, and obtain processed data $\bar{x}$, $\bar{x}''$ and processed mask $\bar{m}$, $\bar{m}'$.

6: $\qquad$ **Forward**: network forward by equation 2, 3, and obtain embedding $z'$, $z$ and reconstruction $\hat{x}'$

7: $\qquad$ **Loss**: Compute loss by equation 5, in which $L_{clu}$ is ignored.

8: $\qquad$ **Backward**: Loss backward and update model parameters $\theta_E$ and $\theta_D$.

9: $\qquad$ iteration = iteration + 1

10: $\quad$ **end while**

11: $\quad$ **if** $epoch \bmod E_s = 0$ & $epoch > E_c$ **then**

12: $\qquad$ Save checkpoint

13: $\quad$ **end if**

14: $\quad$ epoch = epoch + 1

15: **end while**

16: Gather all saved checkpoints. $\qquad\qquad\qquad\qquad$ ▷ Clustering Module Initialization

17: Compute unsupervised metric scores $s$ for all checkpoints by equation 24.

18: Load checkpoint $argmin(s)$, and initialize cluster centers $\mu$.

19: **while** $epoch < E_p + E_j$ **do** $\qquad\qquad\qquad\qquad\qquad\qquad$ ▷ Stage 2: Joint Training

20: $\quad$ iteration = 0

21: $\quad$ **while** $iteration < I_e$ **do**

22: $\qquad$ **Pre-process**: KNN Imputation and Data Augmentation by equation 19 and 20, and obtain processed data $\bar{x}$, $\bar{x}''$ and processed mask $\bar{m}$, $\bar{m}'$.

23: $\qquad$ **Forward**: network forward by equation 2, 3, and obtain embedding $z'$, $z$, reconstruction $\hat{x}'$, and clustering result $c'$

24: $\qquad$ **Loss**: Compute loss by equation 5.

25: $\qquad$ **Backward**: Loss backward and update model parameters $\theta_E$, $\theta_D$, and $\theta_C$.

26: $\qquad$ iteration = iteration + 1

27: $\quad$ **end while**

28: $\quad$ epoch = epoch + 1

29: **end while**

**Output:** Model parameters $\theta_E$, $\theta_D$, $\theta_C$, final clustering result $c$

## C    EXPERIMENTS APPENDIX

### C.1    IMPLEMENTATION DETAILS

**Hyperparameter Settings.** We set most of the hyperparameters empirically with grid search, and the same setting is used for all experiments if not specifically mentioned. $\alpha$ and $\beta$ in equation 11 is set to 20 and -3 respectively. $\gamma$ in equation 15 is set to -10. The hyperparameter $k$ in KNN Imputation is set to 4. The data augmentation hyperparameter in equation 20 $\phi_2$, and $\phi_3$ are fixed at 0.05, while $\phi_1$ in equation 19 which controls view dropout possibility is set to be growing with the actual missing rate of the dataset, defined as,

$$\phi_1 = \epsilon + (1 - \epsilon) \times (1 - \frac{\sum_{i=0}^{N} \sum_{j=0}^{V} M_{ij}}{N \times V})^2 \tag{25}$$

in which we set $\epsilon = 0.15$. The loss weight hyperparameters $\lambda_1$ and $\lambda_2$ are set to 0.001 and 0.1 respectively. $\omega$ in equation 24 is set to 5. The embedding dimension $d_e$ is set to 256. Batch size $B$ is set to 64 for both training and testing and the learning rate is fixed at 3e-4 throughout training. A small weight decay of 4e-5 is used for less over-fitting. The training epoch parameter $E_p$ (Appendix B.8)is set to 100, 100, 15, and 50 respectively for the four datasets in Table 1 to maintain roughly the same training iteration. $E_c = 0.7E_p$ is set for all datasets. As for $E_j$, we found that training with DEC-based loss on some datasets (ALOI_Deep and Scene15 in this paper) can diverge, possibly due to imbalanced cluster size. For these datasets, we simply skip the second stage's joint training, i.e., $E_j = 0$, while for other datasets $E_j = E_p$.

**Design details.** PReLU (He et al., 2015) is used as the activation function in the VDE and the Decoder. Dropout is not used in any modules of our network. Agglomerative clustering with "ward" linkage is used for initializing cluster centers in the Clustering Module.

### C.2    DATASETS AND EXPERIMENTS SETTING

Our chosen datasets vary in views (2–6), clusters (7–100), samples (1400–10800), and feature types (deep feature/hand-crafted feature), providing a comprehensive evaluation of approaches. Two parameters missing number $m_n$ and missing rate $m_r$ are defined to control the missing conditions. We first select $N \times m_r$ samples as incomplete samples, then randomly select $m_n$ views of each incomplete sample as missing views. We fix $m_n$ and vary $m_r$ in our experiments, $m_n$ are fixed at 4, 3, 2, 1 for the 4 datasets respectively. Importantly, it should be noted that within the same set of experiments, we ensured that the input data and missing indicator matrix remained consistent across different methods, ensuring fair comparisons. For comparison with state-of-the-art methods in Table 2 we reproduce the results with their published code, while for experiments in Table 6 we align our dataset settings with MPC (Liu et al., 2022) and directly cite their results. Several prior works are difficult to adapt to different numbers of views, which could hinder real applications. For instance, Completer (Lin et al., 2021) and SURE (Yang et al., 2023b) are only capable with 2-view datasets, and DCP (Lin et al., 2023) can only handle datasets with fewer than 4 views without extensively modifying the code. We randomly select views when the dataset has more views than required.

### C.3    COMPARISON WITH STATE-OF-THE-ART METHODS

Table 4: The statistic of 2 datasets with textual features.

| Name | Views | Clusters | Samples | Dimensions |
|---|---|---|---|---|
| BDGP (Cai et al., 2012; Tang & Liu, 2022) | 2 | 5 | 2500 | 1750/79 |
| Reuters (Amini et al., 2009; Yang et al., 2023b) | 2 | 6 | 18758 | 10/10 |

**Comparison on textual datasets.** The experiments in Table 2 are conducted mainly on image features, so another experiment in Table 5 is conducted to examine the model's performance on textual datasets (Table 4). BDGP (Cai et al., 2012; Tang & Liu, 2022) is a multi-modal dataset with 1750-d visual features and 79-d textual features as two views. Reuters (Amini et al., 2009; Yang et al., 2023b) is a multilingual dataset, English and French are used as two views. As shown in Table 2, our approach still achieves optimal or suboptimal results with textual features, proving its effectiveness.

Table 5: Comparison of our method with state-of-the-art approaches on 2 benchmark datasets with textual input. A fixed missing rate $m_r = 0.5$ is applied to all datasets. The best result is highlighted in **bold** while the suboptimal is underlined.

| Datasets | BDGP | | | Reuters | | |
|---|---|---|---|---|---|---|
| Metrics | Acc(%) | NMI(%) | ARI(%) | Acc(%) | NMI(%) | ARI(%) |
| Completer (Lin et al., 2021) | $58.37 \pm 4.87$ | $48.52 \pm 4.34$ | $25.13 \pm 6.71$ | $40.30 \pm 7.32$ | $22.86 \pm 6.44$ | $10.53 \pm 8.25$ |
| DSIMVC (Tang & Liu, 2022) | $\mathbf{95.71 \pm 0.21}$ | $\mathbf{87.08 \pm 0.54}$ | $\mathbf{89.62 \pm 0.51}$ | $48.39 \pm 2.92$ | $\mathbf{31.88 \pm 2.27}$ | $\mathbf{26.04 \pm 2.27}$ |
| SURE (Yang et al., 2023b) | $63.27 \pm 7.55$ | $41.35 \pm 7.70$ | $36.51 \pm 8.87$ | $48.63 \pm 3.56$ | $27.73 \pm 2.46$ | $22.46 \pm 1.59$ |
| DCP (Lin et al., 2023) | $55.82 \pm 7.02$ | $44.56 \pm 8.92$ | $20.08 \pm 11.71$ | $\underline{39.02 \pm 3.16}$ | $22.47 \pm 4.19$ | $6.92 \pm 3.34$ |
| CPSPAN (Jin et al., 2023) | $81.40 \pm 8.64$ | $66.19 \pm 6.61$ | $64.71 \pm 9.89$ | $39.35 \pm 2.13$ | $14.47 \pm 2.01$ | $12.37 \pm 1.79$ |
| RecFormer (Liu et al., 2023) | $49.76 \pm 3.49$ | $38.62 \pm 3.30$ | $19.01 \pm 2.68$ | $39.70 \pm 5.14$ | $17.27 \pm 2.83$ | $14.82 \pm 3.19$ |
| URRL-IMVC (ours) | $\underline{92.52 \pm 5.71}$ | $\underline{82.29 \pm 5.66}$ | $\underline{84.26 \pm 7.57}$ | $\mathbf{49.91 \pm 2.44}$ | $\underline{29.49 \pm 1.38}$ | $\underline{25.71 \pm 2.08}$ |

Table 6: Comparison with latest IMVC methods. All experimental setups were kept consistent with MPC, so we directly inherited results from MPC's paper. "Fp" and "Fb" represent Pairwise and Bcubed F-measure respectively. The best result is in **bold** and suboptimal underlined.

| | Dataset | Handwritten (Duin, 2023) | | | | 100Leaves (Mallah et al., 2013) | | | | Humbi240 (Yu et al., 2020) | | | |
|---|---|---|---|---|---|---|---|---|---|---|---|---|---|
| Task | Method | Fp | Fb | NMI | ARI | Fp | Fb | NMI | ARI | Fp | Fb | NMI | ARI |
| MVC | OSLF (Zhang et al., 2021) | 76.4 | 76.49 | 76.51 | 73.79 | 65.55 | 69.59 | 87.68 | 65.2 | 90.35 | 93.62 | 98.2 | 90.31 |
| | EEIMC (Liu et al., 2021) | 75.86 | 76.52 | 78.28 | 73.17 | 74.1 | 77.53 | 91.18 | 73.84 | 91.45 | 94.45 | 98.54 | 91.41 |
| | UEAF (Wen et al., 2019) | 82.57 | 82.45 | 83 | 79.91 | 64.54 | 72.81 | 89.18 | 64.16 | 86.36 | 90.36 | 97.11 | 86.3 |
| | PIC (Wang et al., 2019a) | 80.84 | 81.14 | 83.26 | 78.72 | 78.04 | 81.49 | 92.76 | 77.82 | 94.34 | 96.29 | 98.95 | 94.32 |
| | MPC (Liu et al., 2022) | $\underline{90.17}$ | $\underline{89.78}$ | $\underline{89.77}$ | $\underline{89.15}$ | $\underline{84.18}$ | $\underline{85.65}$ | $\mathbf{94.40}$ | $\mathbf{84.04}$ | $\underline{95.49}$ | $\underline{97.03}$ | $\underline{99.07}$ | $\underline{95.47}$ |
| | URRL-IMVC (Ours) | $\mathbf{97.66}$ | $\mathbf{95.45}$ | $\mathbf{94.74}$ | $\mathbf{94.79}$ | $\mathbf{85.09}$ | $\mathbf{85.75}$ | $\underline{94.33}$ | $\underline{81.74}$ | $\mathbf{98.52}$ | $\mathbf{98.51}$ | $\mathbf{99.50}$ | $\mathbf{98.21}$ |
| IMVC | OSLF (Zhang et al., 2021) | 64.56 | 65.88 | 69.75 | 60.48 | 33.86 | 39.04 | 71.84 | 33.19 | 70.72 | 73.40 | 89.41 | 70.59 |
| | EEIMC (Liu et al., 2021) | 78.26 | 78.79 | 79.53 | 75.85 | 52.65 | 56.74 | 81.11 | 52.18 | 80.94 | 86.24 | 94.84 | 80.86 |
| | UEAF (Wen et al., 2019) | 81.54 | 81.88 | 82.39 | 79.49 | 38.47 | 45.87 | 75.62 | 37.82 | 86.04 | 89.96 | $\underline{96.81}$ | 85.98 |
| | PIC (Wang et al., 2019a) | 78.46 | 79.32 | 81.34 | 76.04 | 50.79 | 55.61 | 80.72 | 50.30 | 83.30 | 85.74 | 94.64 | 83.23 |
| | MPC (Liu et al., 2022) | $\underline{90.17}$ | $\underline{89.78}$ | $\underline{89.77}$ | $\underline{89.15}$ | $\underline{58.31}$ | $\underline{61.19}$ | $\underline{83.39}$ | $\underline{57.94}$ | $\underline{90.10}$ | $\underline{91.56}$ | 96.53 | $\underline{90.06}$ |
| | URRL-IMVC (Ours) | $\mathbf{97.80}$ | $\mathbf{95.73}$ | $\mathbf{95.05}$ | $\mathbf{95.12}$ | $\mathbf{73.07}$ | $\mathbf{67.58}$ | $\mathbf{86.21}$ | $\mathbf{62.08}$ | $\mathbf{95.47}$ | $\mathbf{93.56}$ | $\mathbf{97.14}$ | $\mathbf{92.33}$ |

**Comparison with traditional IMVC methods.** Besides the DIMVC approaches, we also compared our method with traditional IMVC approaches, the results are recorded in Table 6. The proposed URRL-IMVC achieved consistently superior performance for the IMVC task regardless of datasets and metrics, while for the MVC task, it achieved the best performance on Handwritten and the Humbi240 dataset, and being comparable with MPC on 100Leaves. Generally speaking, our proposed framework also achieved state-of-the-art results compared with traditional IMVC approaches.

Table 7: Comparison between our approach and cross-view contrastive learning-based approach (CPSPAN) on Caltech101-7 dataset with a different number of views. The best results of CPSPAN are achieved with 4 views, while with 5 views for our approach.

| Views | CPSPAN (Jin et al., 2023) | | | URRL-IMVC (ours) | | |
|---|---|---|---|---|---|---|
| | Acc(%) | NMI(%) | ARI(%) | Acc(%) | NMI(%) | ARI(%) |
| 2 | $50.88 \pm 1.87$ | $45.27 \pm 2.50$ | $35.79 \pm 2.25$ | $58.36 \pm 3.01$ | $47.16 \pm 2.50$ | $39.40 \pm 2.71$ |
| 3 | $73.17 \pm 4.27$ | $61.40 \pm 4.29$ | $55.37 \pm 5.46$ | $77.60 \pm 0.88$ | $67.61 \pm 0.98$ | $63.97 \pm 1.33$ |
| 4 | $\mathbf{84.89 \pm 2.15}$ | $\mathbf{75.37 \pm 2.45}$ | $\mathbf{71.79 \pm 3.26}$ | $91.73 \pm 0.47$ | $\underline{83.57 \pm 0.68}$ | $83.26 \pm 0.76$ |
| 5 | $\underline{77.62 \pm 4.74}$ | $\underline{69.70 \pm 4.04}$ | $\underline{63.23 \pm 5.51}$ | $\mathbf{92.95 \pm 2.60}$ | $\mathbf{86.29 \pm 1.76}$ | $\mathbf{86.02 \pm 2.91}$ |

**Comparison with a different number of views.** As we mentioned in the introduction, the effectiveness of the cross-view contrastive learning strategy diminishes due to less overlapped information between views. Observing from Table 7, adding more views may harm the clustering performance of the cross-view contrastive learning-based approach, proving this point of view, and also being consistent with the theoretical analysis from (Trosten et al., 2023). On the other hand, our approach stably benefits from more views in the dataset, overcoming this drawback.

Table 8: Comparison of model parameters and computational costs (Multiply-Adds, MACs) with state-of-the-art DIMVC approaches.

| Approach | MACs (M) | Parameters (M) |
|---|---|---|
| Completer (Lin et al., 2021) | 8.98 | 6.82 |
| DSIMVC (Tang & Liu, 2022) | 9.89 | 9.33 |
| SURE (Yang et al., 2023b) | 8.24 | 8.24 |
| DCP (Lin et al., 2023) | 9.02 | 9.01 |
| CPSPAN (Jin et al., 2023) | 4.92 | 4.93 |
| RecFormer (Liu et al., 2023) | 12.08 | 6.09 |
| URRL-IMVC (ours) | **1.54** | **1.01** |

**Comparison of model parameters and computational costs.** We calculated single forward statistics with the "thop" library[1] on the Scene-15 dataset, the results are depicted in Table 8. Except for the state-of-the-art performance, our model also enjoys fewer parameters and computational costs thanks to our efficient design. Like prior works, the network hyperparameters of our model are not carefully chosen, and the same network is used for all datasets. Thus, there might still be room for improvement.

## C.4 DISCUSSION ON SCENE15

Our approach achieved 41.69/40.88/23.93 for Acc/NMI/ARI on the Scene15 dataset, with slightly lower NMI than Completer. So we conducted experiments searching for better performance and found that training the network to 5 times the original iteration results in 43.93/43.31/26.43 which is remarkably better than the original setting. However, not all datasets benefit from 5 times of training. For instance, on Handwritten and Caltech101-7, 5 times of training results in worse performance possibly due to auto-encoder overfitting. Currently, we have found four datasets that benefit from more iterations of training, Scene15, Humbi240, BDGP, and 100Leaves (2 views). A hypothesis for this phenomenon is that this characteristic is related to the number of views, as these three datasets are all 2-view datasets. But it remains to be proved by future work, so in the paper, we still report the performance without 5 times of training.

## C.5 ABLATION ON DESIGN DETAILS

Table 9: Ablation test on the output choice of VDE and NDE. "Mean" represents using the average of all output vectors from the Transformer as output. The $1^{st}$ represents using the first output vector, $2^{nd}$ represents the second output vector, and so on. "Concat+Linear" represents first concatenating the output vectors and then using a linear layer to map the new vector to the desired dimension.

| Output/Module | NDE | VDE |
|---|---|---|
| $1^{st}$ | **93.36±0.89** | 89.73±3.26 |
| $2^{nd}$ | 91.87±1.01 | 88.75±3.56 |
| $3^{rd}$ | 90.47±3.15 | 87.23±3.67 |
| $4^{th}$ | 89.36±4.33 | 89.94±4.15 |
| $5^{th}$ | - | 87.14±3.88 |
| Mean | 90.09±3.61 | **93.36±0.98** |
| Concat+Linear | 83.23±7.16 | 91.13±3.38 |

**Ablation on Output Choice.** We conducted the ablation test in Table 9 to find the best output choice of both the Neighbor Dimensional Encoder (NDE, section 3.1.1) and the View Dimensional Encoder (VDE, section 3.1.2). It can be observed that choosing the first vector of the Transformer output sequence significantly outperforms other choices, and using the latter output vectors results in worse and worse performance. It is consistent with our point of view that NDE needs a bias on

---

[1]https://github.com/Lyken17/pytorch-OpCounter

the most confident input (the center sample or the nearest neighbor), and further neighbors contain more noise to harm the final performance. For VDE the situation is different, using the average of all output vectors outperforms other choices, which is consistent with our point of view that VDE needs to be unbiased. Concatenation with linear layer does not perform well in both Encoders, possibly due to lack of supervision.

Table 10: Ablation test on traditional clustering method for initialization. K-means and Agglomerative clustering with "ward" linkage are tested. Agglomerative clustering is much better both on initialization and final clustering results thanks to its stability.

| Cluster Method | K-means | Agglomerative |
|---|---|---|
| Initialize | $84.84 \pm 6.08$ | $\mathbf{90.38 \pm 1.46}$ |
| Final | $88.34 \pm 6.33$ | $\mathbf{93.36 \pm 0.98}$ |

**Ablation on Clustering Method.** As shown in Table 10, we tried two kinds of traditional clustering methods for Clustering Module initialization (Appendix B.4), K-means and Agglomerative clustering with "ward" linkage. Agglomerative clustering achieved much better results mainly due to its stability in initialization. As we mentioned earlier in the paper, DEC-based clustering training is sensitive to initialization, so a stable initialization can result in stable final clustering results.

Table 11: Ablation test on positional encoding (PE) for NDE. Cosine distance-based and learnable PE achieved the best results, while sinusoidal PE had negative effects.

| Type | Cosine | Euclidean | Learnable | Sinusoidal | None |
|---|---|---|---|---|---|
| Concat | $\mathbf{93.36 \pm 0.98}$ | $92.73 \pm 0.45$ | $\underline{93.20 \pm 0.59}$ | $90.91 \pm 2.40$ | $92.63 \pm 2.30$ |
| Add | - | - | $90.96 \pm 3.60$ | $92.29 \pm 2.96$ | - |

**Ablation on Positional Encoding.** We conducted an ablation study on the design of Positional Encoding (PE) in NDE (section 3.1.1), the results are recorded in Table 11. Cosine distance and Learnable PE with concatenation achieved the best results, while Euclidean distance PE achieved secondary results but with the best stability. Sinusoidal PE had negative effects, especially when concatenated with input. The result is consistent with the ablation study in (Nguyen et al., 2021), while our PE generally has a smaller impact on the final result. The difference can come from our other designs or the supervision.

Table 12: Ablation study on extra missing view recovery network and robustness learning. "RL" represents the Robustness Learning in URRL-IMVC that consists of augmentation and robustness loss. Adding an extra network 1 for missing view recovery only results in comparable performance compared with our one network setting, while lacking Robustness Learning can cause a significant performance drop. The phenomenon supports our point of view that directly learning robust embedding is more efficient than explicitly recovering missing views.

| Network 1 | Network 2 | Acc(%) | NMI(%) | ARI(%) |
|---|---|---|---|---|
| - | with RL | $\mathbf{92.95 \pm 2.60}$ | $\mathbf{86.29 \pm 1.76}$ | $\mathbf{86.02 \pm 2.91}$ |
| with RL | with RL | $\underline{92.93 \pm 2.64}$ | $\underline{85.93 \pm 1.74}$ | $\underline{84.98 \pm 2.84}$ |
| with RL | w/o RL | $89.65 \pm 4.65$ | $83.34 \pm 3.66$ | $81.72 \pm 5.38$ |
| w/o RL | w/o RL | $87.23 \pm 12.06$ | $82.28 \pm 6.62$ | $79.96 \pm 11.12$ |

**Ablation on Missing View Recovery and Robust Learning.** Our results may raise the question of whether combining our network and missing view recovery could achieve better results. Thus, we conducted an ablation study to explore whether adding an additional missing view recovery network could help. We followed the framework in RecFormer (Liu et al., 2023) and used two identical URRL-IMVC networks, with the first network learning to recover the missing views and the second network doing complete multi-view clustering. As the results in Table 12 show, with our robustness learning strategy, comparable results are achieved with the extra recovery network. Indicating the robustness learning strategy can replace the role of missing view recovery and thus,

save the computational costs and parameters of the extra recovery network. On the other hand, removing the robust learning in both networks can degrade the clustering performance and increase instability, indicating that missing views recovered with simple strategies might be unreliable. The phenomenon supports our point of view in the Introduction 1.

## C.6 Ablation on Hyperparameters

Table 13: We conducted ablation tests on the two hyperparameters that control the balance between three losses. The best result can be found with $\lambda_1 = 0.001$ and $\lambda_2 = 0.1$.

| Loss Hyp | 0.0001 | 0.001 | 0.01 | 0.1 | 1 |
|---|---|---|---|---|---|
| $\lambda_1 (\lambda_2 = 0.1)$ | $92.45 \pm 2.49$ | $\mathbf{93.36 \pm 0.98}$ | $92.47 \pm 0.68$ | $85.37 \pm 4.31$ | $64.14 \pm 3.13$ |
| $\lambda_2 (\lambda_1 = 0.001)$ | $90.19 \pm 1.57$ | $90.23 \pm 1.46$ | $89.95 \pm 1.21$ | $\mathbf{93.36 \pm 0.98}$ | $14.30 \pm 0.02$ |

**Ablation on Loss Weights.** The ablation on loss weights (equation 5) can be found in Table 13. The weight for embedding robustness loss $\lambda_1$ is best set to 0.001, which is also the smallest of the three losses as it actually plays an auxiliary role during training. We found that this loss will automatically decrease with the decrease of reconstruction loss, which is consistent with the intuition that learning to recover dropped-out views shares nearly the same objective as learning to minimize the distance between augmented and original embedding. The experiments also show that a large $\lambda_1$ will reduce the final performance, as the network focuses too much on augmentation and ignores the inherent structure of data. The best weight for clustering loss $\lambda_2$ is 0.1. The performance shows a leaping change near 0.1 and achieves the best performance, while the training collapses when it passes 0.1.

Table 14: The ablation study on view dropout augmentation probability $\phi_1$ from equation 19. We use grid search to determine the best value range of $\phi_1$ under different missing rates $m_r$ and try to design a mapping function from the actual missing rate to the desired $\phi_1$. Generally speaking, a larger missing rate requires a larger view dropout probability for augmentation, the $\phi_1$ value from the designed mapping function, equation 25, is listed in the last column of the table.

| Parameter | $\phi_1 = 0$ | $\phi_1 = 0.15$ | $\phi_1 = 0.3$ | $\phi_1 = 0.45$ | $\phi_1 = 0.6$ | Equation 25 |
|---|---|---|---|---|---|---|
| $m_r = 0.00$ | $89.30 \pm 1.86$ | $\underline{89.50 \pm 1.77}$ | $88.91 \pm 1.89$ | $\mathbf{89.71 \pm 1.19}$ | $86.94 \pm 2.12$ | $\phi_1 = 0.15$ |
| $m_r = 0.25$ | $86.15 \pm 2.42$ | $\underline{88.12 \pm 1.90}$ | $\mathbf{88.56 \pm 1.55}$ | $86.51 \pm 3.33$ | $83.57 \pm 4.74$ | $\phi_1 = 0.17$ |
| $m_r = 0.50$ | $83.35 \pm 1.94$ | $\mathbf{86.12 \pm 1.57}$ | $\underline{85.61 \pm 2.08}$ | $83.25 \pm 3.86$ | $82.25 \pm 4.82$ | $\phi_1 = 0.23$ |
| $m_r = 0.75$ | $81.41 \pm 2.79$ | $\underline{81.78 \pm 3.86}$ | $\mathbf{82.89 \pm 4.51}$ | $80.82 \pm 3.58$ | $77.17 \pm 5.63$ | $\phi_1 = 0.32$ |
| $m_r = 1.00$ | $76.31 \pm 3.00$ | $77.53 \pm 3.63$ | $77.02 \pm 3.91$ | $\mathbf{80.21 \pm 2.94}$ | $\underline{79.19 \pm 3.67}$ | $\phi_1 = 0.46$ |

**Ablation on View Dropout Probability** $\phi_1$**.** We conducted a grid search to determine the best value range of view dropout augmentation probability $\phi_1$ under different missing rates $m_r$, and the results are shown in Table 14. For view complete condition, $\phi_1 < 0.6$ have similar performance, while for view incomplete condition, the desired $\phi_1$ ascends as the missing rate $m_r$ increases. According to this observation, we designed the mapping function in equation 25 to follow this ascending trend, and its value is listed in the last column of the table.

Table 15: Ablation test on the hyperparameter $k$ in KNN. The result is unimodal with the best $k = 4$. Larger $k$ values tend to provide more stable results (smaller standard deviation).

| $k = 1$ | $k = 2$ | $k = 4$ | $k = 8$ | $k = 16$ |
|---|---|---|---|---|
| $83.84 \pm 3.38$ | $84.84 \pm 2.82$ | $\mathbf{87.31 \pm 2.01}$ | $\underline{87.00 \pm 2.46}$ | $86.01 \pm 1.28$ |

**Ablation on hyperparameter** $k$ **for KNN.** We conduct an ablation study on $k$ in KNN Imputation (Appendix B.5) to examine its effect. A large increment can be observed comparing $k = 4$ with $k = 1$. However, the performance starts to drop as $k > 4$, which we infer can be caused by the noise brought by further neighbors. On the other hand, larger $k$ also seems to benefit the stability of clustering.

Table 16: Ablation test on network hyperparameters. We vary Transformer block numbers in NDE and VDE, the Feed Forward Network (FFN) dimension in NDE, VDE, and Decoder, and the unified embedding dimension for this test. We report computational cost (Multiply-Adds, MACs), network parameters, and Accuracy on the Caltech101-7 dataset as results. As observed, larger networks do not necessarily yield better clustering performance, and the best results can be achieved with 2 Transformer blocks and 256 as FFN and embedding dimensions.

| Transformer Blocks | FFN Dimension | Embedding Dimension | MACs | Parameters | Accuracy |
|---|---|---|---|---|---|
| 1 | 128 | 128 | 29.73 | 12.07 | $91.86 \pm 1.80$ |
| 2 | 128 | 128 | 33.65 | 13.03 | $91.29 \pm 3.24$ |
| 2 | 256 | 128 | 43.04 | 16.69 | $90.33 \pm 5.26$ |
| 2 | 256 | 256 | 43.98 | 17.13 | $\mathbf{92.95 \pm 2.60}$ |
| 4 | 256 | 256 | 60.08 | 21.07 | $\underline{91.92 \pm 2.57}$ |
| 4 | 512 | 256 | 97.05 | 34.14 | $89.31 \pm 5.11$ |
| 4 | 512 | 512 | 103.34 | 36.39 | $90.02 \pm 3.29$ |

**Ablation on network hyperparameters.** Though it is not the focus of most prior works, we also conduct an ablation experiment (Table 16) on network hyperparameters to study the influence of model size. It can be observed that larger models do not necessarily yield better clustering accuracy, the best result is achieved with a balanced setting with 2 Transformer blocks in the NDE and VDE each, while FFN and embedding dimensions set to 256. A reasonable explanation for this phenomenon can be, that small models fail to learn the complex fusion mapping between raw data and unified embedding, while large models can easily overfit and weaken the relation between latent space and data space.

## C.7 VISUALIZATION

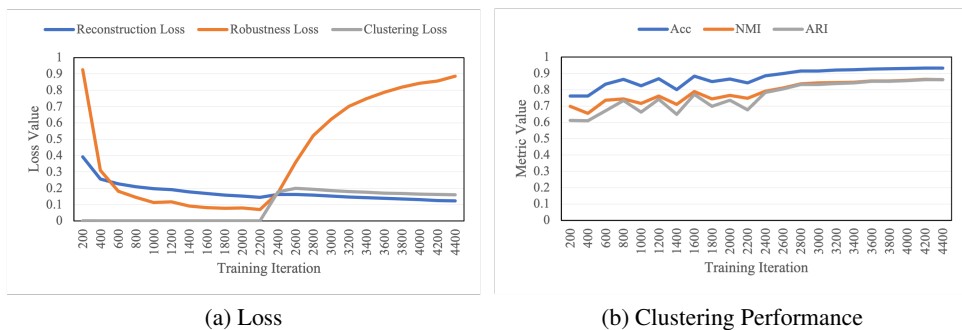

(a) Loss          (b) Clustering Performance

Figure 6: Visualization of loss and clustering performance throughout the training process.

**Visualization of Loss and Performance.** We visualized the three loss terms and clustering performance throughout one training process in Figure 6. In the first stage of training, both reconstruction loss and robustness loss drop. In the second stage, however, reconstruction loss continues to drop while robustness loss is sacrificed for clustering loss. As for clustering performance, in the first stage, the clustering performance vibrates due to a lack of clustering-oriented supervision, and it is stabilized in the second stage.

