# OpenReview forum: "URRL-IMVC: Unified and Robust Representation Learning for Incomplete Multi-View Clustering"
_ICLR.cc/2024/Conference — Submitted to ICLR 2024_

### Official Review · Reviewer_xehP · 2023-10-20

**Soundness:** 2 fair
**Presentation:** 2 fair
**Contribution:** 2 fair
**Rating:** 5
**Confidence:** 3

**Summary:**

This paper learns a unified embedding that is robust to view missing conditions by integrating information from multiple views and neighboring samples. Firstly, to overcome the limitations of cross-view contrastive learning, URRL-IMVC incorporates an attention-based auto-encoder framework to fuse multi-view information and generate unified embeddings. Secondly, URRL-IMVC directly enhances the robustness of the unified embedding against view-missing conditions through KNN imputation and data augmentation techniques, eliminating the need for explicit missing view recovery. Finally, incremental improvements are introduced to further enhance the overall performance.

**Strengths:**

1. The originality, quality, and significance of this paper are supported by the proposed unified representation learning framework that efficiently fuses both multiview and neighborhood information, allowing for better capturing of consensus and complementary information.

2. The clarity of this paper is clear based on the framework figure and the corresponding illustrations.

**Weaknesses:**

1. The biggest problem of this paper is the limited novelty in formulation of URRL-IMVC，which learns a unified embedding that captures the comprehensive representation. The differences between URRL-IMVC and the closely related works can be analyzed from different aspects.

2. The strategies including KNN imputation and data augmentation should be stated in details. Then the process of directly learning a robust representation capable of handling view-missing conditions without explicit missing view recovery is easily understood by the readers.

3. In the experiments, the compared methods are not enough and the more datasets can be added, i.e., Table 2.

4. The convergence analysis can be added in the experiment, which can be adopted to better the loss function.

**Questions:**

Why the visualization for 4400 iteration is not significantly improved compared with 2400 iteration in Figure 4 for the experiment?

---

> ### Author Response · Authors · 2023-11-20
> **Response to Reviewer xehP #1**
>
> ***W1: The biggest problem of this paper is the limited novelty in formulation of URRL-IMVC，which learns a unified embedding that captures the comprehensive representation. The differences between URRL-IMVC and the closely related works can be analyzed from different aspects.***
>
> **R1:** The key contribution of our paper lies in the novel approach of directly learning a unified and robust representation for the IMVC task.
>
> **Unified**: In the DIMVC field, the prevailing framework is cross-view contrastive learning [2] [3] [4] [5]. However, this design primarily emphasizes the consensus information in multi-view data while neglecting the valuable complementary information. Therefore, our approach goes beyond the cross-view contrastive framework and instead learns a unified representation using a meticulously designed attention-based network that effectively leverages complementary information.
>
> **Robust**: The primary approach for handling missing views in the DIMVC field is to first recover the missing views and then perform complete multi-view clustering [1] [2] [3] [4] [5]. However, the recovered views may be unreliable, and computational resources are wasted on repeatedly mapping between data and latent space [1]. To address this issue, we propose directly learning a latent embedding that is robust to view missing, without explicitly recovering the missing data. We have devised data augmentation and KNN imputation as preprocessing techniques, which, along with the unified representation learning framework, facilitate efficient and robust representation learning.
>
> Overall, our paper introduces a novel approach that directly learns a unified and robust representation for the IMVC task, surpassing the limitations posed by existing approaches in the field.
>
> ***W2: The strategies including KNN imputation and data augmentation should be stated in details. Then the process of directly learning a robust representation capable of handling view-missing conditions without explicit missing view recovery is easily understood by the readers.***
>
> **R2:** A detailed description and formulation of the KNN imputation strategy is provided in Appendix B.5 and the data augmentation strategy in Appendix B.6. Given the length limitation of the conference paper (9 pages), we have to include most of the detailed formulations in the Appendix to ensure the paper's completeness. Additionally, in this revised version, we have added Algorithm 1 to describe the KNN imputation strategy, which we hope will facilitate easier interpretation. Regarding data augmentation, we believe that the formulation in equations 19 and 20 already provides sufficient detail.

---

> ### Author Response · Authors · 2023-11-20
> **Response to Reviewer xehP #2**
>
> ***W3: In the experiments, the compared methods are not enough and the more datasets can be added, i.e., Table 2.***
>
> **R3:** In response to this concern, an additional comparison with IMVC methods that do not use deep neural networks has already been included in the previous version paper. It can be found in Table 6, Appendix C.3, and includes results from two extra datasets and five additional methods. Additionally, based on the suggestions from the reviewers, we have included two textual datasets (Table 4) for comparison in this revised version. The corresponding results are presented in Table 5 and below. On both datasets, our approach achieved state-of-the-art performance.
>
> | Datase\Metrics   | BDGP Acc(\%)                 | BDGP NMI(\%)                 | BDGP ARI(\%)                 | Reuters Acc(\%)              | Reuters NMI(\%)              | Reuters ARI(\%)              |
> | ------------------ | ------------------------------ | ------------------------------ | ------------------------------ | ------------------------------ | ------------------------------ | ------------------------------ |
> | Completer        | 58.37 $\pm$ 4.87              | 48.52 $\pm$ 4.34              | 25.13 $\pm$ 6.71              | 40.30 $\pm$ 7.32              | 22.86 $\pm$ 6.44              | 10.53 $\pm$ 8.25              |
> | DSIMVC           | **95.71 $\pm$ 0.21**         | **87.08 $\pm$ 0.54**         | **89.62 $\pm$ 0.51**         | 48.39 $\pm$ 2.92              | **31.88 $\pm$ 2.27**         | **26.04 $\pm$ 2.27**         |
> | SURE             | 63.27 $\pm$ 7.55              | 41.35 $\pm$ 7.70              | 36.51 $\pm$ 8.87              | $\underline{48.63 \pm 3.56}$ | 27.73 $\pm$ 2.46              | 22.46 $\pm$ 1.59              |
> | DCP              | 55.82 $\pm$ 7.02              | 44.56 $\pm$ 8.92              | 20.08 $\pm$ 11.71            | 39.02 $\pm$ 3.16              | 22.47 $\pm$ 4.19              | 6.92 $\pm$ 3.34               |
> | CPSPAN           | 81.40 $\pm$ 8.64              | 66.19 $\pm$ 6.61              | 64.71 $\pm$ 9.89              | 39.35 $\pm$ 2.13              | 14.47 $\pm$ 2.01              | 12.37 $\pm$ 1.79              |
> | RecFormer        | 49.76 $\pm$ 3.49              | 38.62 $\pm$ 3.30              | 19.01 $\pm$ 2.68              | 39.70 $\pm$ 5.14              | 17.27 $\pm$ 2.83              | 14.82 $\pm$ 3.19              |
> | URRL-IMVC (ours) | $\underline{92.52 \pm 5.71}$ | $\underline{82.29 \pm 5.66}$ | $\underline{84.26 \pm 7.57}$ | **49.91 $\pm$ 2.44**         | $\underline{29.49 \pm 1.38}$ | $\underline{25.71 \pm 2.08}$ |
>
> ***W4. The convergence analysis can be added in the experiment, which can be adopted to better the loss function.***
>
> **R4:** Thank you for the suggestion. In Figure 6, Appendix C.7, we have included a visualization of the loss and clustering performance throughout the training process. During the first training stage, we observe that the reconstruction and robustness loss converge. In the second stage, while the reconstruction loss continues to decrease, the robustness loss is compromised to prioritize the clustering loss. Furthermore, we have analyzed the clustering performance. During the first stage, the clustering performance exhibits some fluctuations due to the lack of clustering-oriented supervision. However, in the second stage, the clustering performance stabilizes with clustering loss. We believe that this additional analysis provides insights into the convergence behavior of our proposed method.
>
> **Reference**
> - [1]. Chengliang Liu, Jie Wen, Zhihao Wu, Xiaoling Luo, Chao Huang, and Yong Xu. Information recovery-driven deep incomplete multiview clustering network. IEEE Transactions on Neural Networks and Learning Systems, pp. 1–11, 2023. doi: 10.1109/TNNLS.2023.3286918.
> - [2]. Jiaqi Jin, Siwei Wang, Zhibin Dong, Xinwang Liu, and En Zhu. Deep incomplete multi-view clustering with cross-view partial sample and prototype alignment. In Proceedings of the IEEE/CVF Conference on Computer Vision and Pattern Recognition (CVPR), pp. 11600–11609, June 2023.
> - [3]. Mouxing Yang, Yunfan Li, Peng Hu, Jinfeng Bai, Jiancheng Lv, and Xi Peng. Robust multi-view clustering with incomplete information. IEEE Transactions on Pattern Analysis and Machine Intelligence, 45(1):1055–1069, 2023b. doi: 10.1109/TPAMI.2022.3155499.
> - [4]. Huayi Tang and Yong Liu. Deep safe incomplete multi-view clustering: Theorem and algorithm. In Kamalika Chaudhuri, Stefanie Jegelka, Le Song, Csaba Szepesvari, Gang Niu, and Sivan Sabato (eds.), Proceedings of the 39th International Conference on Machine Learning, volume 162 of Proceedings of Machine Learning Research, pp. 21090–21110. PMLR, 17–23 Jul 2022.
> - [5]. Yijie Lin, Yuanbiao Gou, Zitao Liu, Boyun Li, Jiancheng Lv, and Xi Peng. Completer: Incomplete multi-view clustering via contrastive prediction. In Proceedings of the IEEE/CVF Conference on Computer Vision and Pattern Recognition (CVPR), pp. 11174–11183, June 2021.

---

> ### Author Response · Authors · 2023-11-20
> **Response to Reviewer xehP #3**
>
> ***Q1: Why the visualization for 4400 iteration is not significantly improved compared with 2400 iteration in Figure 4 for the experiment?***
>
> **A1:** Our method employs a two-stage training strategy (Algorithm 2, Appendix B.8.), where the first stage focuses on representation learning and the second stage emphasizes clustering learning. The second stage commences at 2200 iterations and continues until 4400 iterations. Similar to many other learning tasks such as classification, the training process exhibits quick convergence initially, which gradually slows down over time. This explains why the accuracy improvement in the first 200 iterations is 2.28%, whereas the last 2000 iterations only result in a 4% increase in accuracy. However, the visualization in Figure 4 clearly demonstrates a substantial enhancement in cluster compactness, validating the effectiveness of our clustering-oriented learning approach. For a more comprehensive understanding of our training process, we suggest referring to the newly added Figure 6.

---

### Official Review · Reviewer_LBwN · 2023-10-30

**Soundness:** 2 fair
**Presentation:** 3 good
**Contribution:** 2 fair
**Rating:** 5
**Confidence:** 4

**Summary:**

This paper introduces URRL-IMVC, an incomplete multi-view clustering method that does not rely on cross-view contrastive learning or missing view recovery. Instead, it leverages the complementarity of information across views by fusing data from two carefully designed encoders. It also eliminates explicit missing view recovery by employing KNN imputation and data augmentation techniques.

**Strengths:**

* The organization of this paper is clear and the motivation is easy to understand.

* Experimental results support the effectiveness of the proposed method.

**Weaknesses:**

* This paper asserts that cross-view contrastive learning may overlook complementary information, and contrasting the unified embedding has the potential to capture a more comprehensive representation. However, there is a lack of both theoretical and experimental evidence to support these claims from either perspective.

* More experiments need to be added to verify the sensitivity of model parameters, such as the setting of k in KNN and the initialization of cluster centers in clustering module.

* The ablation studies on modules are rough. The effectiveness of incremental improvements on each module should be further investigated.

* The related work Section appears to be somewhat concise. Some recent works should be discussed.

**Questions:**

See weakness.

---

> ### Author Response · Authors · 2023-11-20
> **Response to Reviewer LBwN #1**
>
> ***W1: This paper asserts that cross-view contrastive learning may overlook complementary information, and contrasting the unified embedding has the potential to capture a more comprehensive representation. However, there is a lack of both theoretical and experimental evidence to support these claims from either perspective.***
>
> **R1:** The limitations of cross-view contrastive learning are intuitive and have been previously studied in existing literature. It is intuitive to understand that if the contrastive learning objective is achieved, the resulting embeddings from each view would be almost identical, thereby disregarding any view-specific or complementary information. Furthermore, a theoretical analysis conducted by [1] emphasizes that cross-view contrastive alignment can actually reduce the number of distinct clusters in the representation space, and this effect becomes more pronounced with an increasing number of views. Additionally, [2] mentions the inherent conflict between the reconstruction target and the cross-view contrastive target, as the former aims to preserve view-specific information while the latter attempts to ignore it.
>
> In contrast, our unified design overcomed this drawback by learning unified representations. Although currently lacking theoretical evidence due to the black box nature of the unified embedding. However, we have made two experimental observations that provide circumstantial evidence of the preservation and utilization of both consensus and complementary information. Firstly, we noticed in our experiments that our approach outperforms cross-view contrastive learning-based methods on datasets that possess more complementary information. For instance, on the 100Leaves dataset where the two views represent the shape and texture of a leaf, our approach exhibits greater advantages. Secondly, we observed that the clustering performance of cross-view contrastive learning-based approaches declines when additional views are added, as theoretically analyzed in [1]. In contrast, our approach consistently benefits from the inclusion of more views in the dataset, thereby overcoming this drawback. The results of the second experiment have been included in the revised version of the paper as Table 7 in Appendix C.3.
>
> The results from Table 7 are as follows:
> | Views | CPSPAN Acc(\%)               | CPSPAN NMI(\%)               | CPSPAN ARI(\%)               | URRL-IMVC (ours) Acc(\%)     | URRL-IMVC (ours) NMI(\%)     | URRL-IMVC (ours) ARI(\%)     |
> | ------- | ------------------------------ | ------------------------------ | ------------------------------ | ------------------------------ | ------------------------------ | ------------------------------ |
> | 2     | 50.88 $\pm$ 1.87              | 45.27 $\pm$ 2.50              | 35.79 $\pm$ 2.25              | 58.36 $\pm$ 3.01              | 47.16 $\pm$ 2.50              | 39.40 $\pm$ 2.71              |
> | 3     | 73.17 $\pm$ 4.27              | 61.40 $\pm$ 4.29              | 55.37 $\pm$ 5.46              | 77.60 $\pm$ 0.88              | 67.61 $\pm$ 0.98              | 63.97 $\pm$ 1.33              |
> | 4     | **84.89 $\pm$ 2.15**         | **75.37 $\pm$ 2.45**         | **71.79 $\pm$ 3.26**         | $\underline{91.73 \pm 0.47}$ | $\underline{83.57 \pm 0.68}$ | $\underline{83.26 \pm 0.76}$ |
> | 5     | $\underline{77.62 \pm 4.74}$ | $\underline{69.70 \pm 4.04}$ | $\underline{63.23 \pm 5.51}$ | **92.95 $\pm$ 2.60**         | **86.29 $\pm$ 1.76**         | **86.02 $\pm$ 2.91**         |

---

> ### Author Response · Authors · 2023-11-20
> **Response to Reviewer LBwN #2**
>
> ***W2: More experiments need to be added to verify the sensitivity of model parameters, such as the setting of k in KNN and the initialization of cluster centers in clustering module.***
>
> ***W3: The ablation studies on modules are rough. The effectiveness of incremental improvements on each module should be further investigated.***
>
> **R2 & R3:** Thank you for your valuable feedback.
> To address these concerns, we have already conducted extensive ablation studies to evaluate the impact of our design choices, which has been mentioned by Reviewer 3U96. However, due to the limited page length of our conference paper, we were unable to include these studies in the main paper. Instead, we have included them in Appendix C, where readers can find detailed information about our ablation experiments related to both design and hyperparameters. In Appendix C.5, we present the ablation studies focusing on design choices, which demonstrate the effectiveness of our proposed approach, such as the choice of initialization method. In Appendix C.6, we provide ablation studies specifically examining the sensitivity of hyperparameters, such as the setting of k in KNN. To ensure that these important results are not overlooked by readers, we have revised the paper and emphasized cross-references in the main text, directing readers to the relevant sections in the appendix.
>
> Specifically, regarding the initialization of cluster centers in the clustering module, we provided information in Appendix C.1 "Design Details." We used Agglomerative clustering with "ward" linkage, which ensures deterministic results across multiple runs with the same embeddings. To further investigate the impact of initialization methods, we have also conducted an ablation test using K-Means as the initialization method. We found Agglomerative clustering shows better and stabler performance. The results can be found in Table 10 and below.
>
> | Cluster Method | K-means         | Agglomerative        |
> | ---------------- | ----------------- | ---------------------- |
> | Initialize     | 84.84 $\pm$ 6.08 | **90.38 $\pm$ 1.46** |
> | Final          | 88.34 $\pm$ 6.33 | **93.36 $\pm$ 0.98** |
>
> Regarding the setting of k in KNN, the related Ablation study has been presented in Table 15. Near-optimal results can be achieved with k>=4. The result is unimodal with the best k = 4, and larger k values tend to provide more stable results (smaller standard deviation). The results from Table 15 are as follows:
>
> | k=1             | k=2             | k=4                  | k=8                          | k=16            |
> | ----------------- | ----------------- | ---------------------- | ------------------------------ | ----------------- |
> | 83.84 $\pm$ 3.38 | 84.84 $\pm$ 2.82 | **87.31 $\pm$ 2.01** | $\underline{87.00 \pm 2.46}$ | 86.01 $\pm$ 1.28 |
>
> We hope that these experiments will effectively address your concerns about the sensitivity of model parameters and the investigation of incremental improvements on each module.
>
> ***W4: The related work Section appears to be somewhat concise. Some recent works should be discussed.***
>
> **R4:** We acknowledge the importance of discussing recent works in our field. However, due to the length limitation of the paper, we had to move a portion of the related works section to Appendix A.1. This part provides an introduction to recent IMVC works that do not utilize deep neural networks. Additionally, in this revised version, we have included several more recent works in Appendix A.1 to ensure that we cover a broader range of relevant literature.
>
> **Reference**
> - [1]. Daniel J. Trosten, Sigurd Løkse, Robert Jenssen, and Michael C. Kampffmeyer. On the effects of self-supervision and contrastive alignment in deep multi-view clustering. In Proceedings of the IEEE/CVF Conference on Computer Vision and Pattern Recognition (CVPR), pp. 23976–23985, June 2023.
> - [2]. Jie Xu, Huayi Tang, Yazhou Ren, Liang Peng, Xiaofeng Zhu, and Lifang He. Multi-level feature learning for contrastive multi-view clustering. In Proceedings of the IEEE/CVF Conference on Computer Vision and Pattern Recognition (CVPR), pp. 16051–16060, June 2022.

---

### Official Review · Reviewer_umVo · 2023-10-30

**Soundness:** 2 fair
**Presentation:** 2 fair
**Contribution:** 3 good
**Rating:** 5
**Confidence:** 3

**Summary:**

This paper proposes a novel Unified and Robust Representation Learning for Incomplete Multi-View Clustering, which tries to learn a unified embedding that is robust to view missing conditions by integrating information from multiple views and neighboring samples. The method proposed in this paper is without explicit missing view recovery procedure, which is orthogonal to existing missing view recovery-based methods.

**Strengths:**

1. This paper first provides the idea to simultaneously consider the multi-view information fusion and neighborhood information incorporation for clustering under the view-missing conditions.
2. To my knowledge, using an attention-based auto-encoder framework to fuse multi-view information is somewhat novel in multi-view learning.
3. The experimental results are significantly better than former methods.

**Weaknesses:**

1. No evidence for the analysis on the computation cost and unreliability of explicit missing view recovery.
2. The writing of the methodology is not compact and unclear: Some related contents are far away from each other, for example, Eq. (5) and its details.
3. Some essential details are missing, for example, the calculation of KL divergence in Eq. (23) is too abstract to follow.
4. Most of the formulations are postponed to the appendix, making the main text hard to be understood and undermining the clarity.
5. No solid theoretical guarantee is provided for the proposed method.

**Questions:**

1. What is the first output of NDE module in the output choice of the proposed NDE?
2. How does the Siamese Encoder work in Figure 2? Do you mean the architectures of the upper and lower parts are identical with shared parameters?
3. How to determine the level of noise added to the original incomplete multi-view data?
4. Do you consider the private information in each view? In addition to the consensus information, the complementary information is also essential, which has been indicated in this paper. However, I cannot understand what is the mechanism used to leverage the view-specific information in the proposed method. Please clarify this in details.

---

> ### Author Response · Authors · 2023-11-20
> **Response to Reviewer umVo #1**
>
> ***W1: No evidence for the analysis on the computation cost and unreliability of explicit missing view recovery.***
>
> **R1**: Regarding the concern, we have added an ablation study in Table 12 to provide experimental evidence supporting our point of view. In this study, we adopted a similar framework to RecFormer [1] and constructed two identical networks. The first network was used to recover the missing views, while the second network was employed to cluster the recovered complete multi-view data. We achieved comparable results with this framework, which suggests that our robustness learning can effectively replace the role of a missing view recovery network while enjoying lower computational costs.
>
> Furthermore, our experiments have shown that without the robustness objective, the performance of this framework can deteriorate. This observation suggests that missing views recovered with simple strategies may indeed be unreliable. We believe that these findings strengthen our argument and provide empirical support for the analysis of the computation cost and unreliability of explicit missing view recovery.
>
> The results of Table 12 are as follows, in which RL represents Robustness Learning:
> | Network 1 | Network 2 | Acc(\%)                      | NMI(\%)                      | ARI(\%)                      |
> | ----------- | ----------- | ------------------------------ | ------------------------------ | ------------------------------ |
> | -         | with RL   | **92.95 $\pm$ 2.60**         | **86.29 $\pm$ 1.76**         | **86.02 $\pm$ 2.91**         |
> | with RL   | with RL   | $\underline{92.93 \pm 2.64}$ | $\underline{85.93 \pm 1.74}$ | $\underline{84.98 \pm 2.84}$ |
> | with RL   | w/o RL    | 89.65 $\pm$ 4.65              | 83.34 $\pm$ 3.66              | 81.72 $\pm$ 5.38              |
> | w/o RL    | w/o RL    | 87.23 $\pm$ 12.06             | 82.28 $\pm$ 6.62              | 79.96 $\pm$ 11.12             |
>
> ***W2: The writing of the methodology is not compact and unclear: Some related contents are far away from each other, for example, Eq. (5) and its details.***
>
> ***W3: Some essential details are missing, for example, the calculation of KL divergence in Eq. (23) is too abstract to follow.***
>
> ***W4: Most of the formulations are postponed to the appendix, making the main text hard to be understood and undermining the clarity.***
>
> **R2 & R3 & R4:** Given the paper length limitation of this conference, we had to move some of the detailed formulations and ablation studies to the Appendix. This may have caused difficulty in finding relevant equations or contents. In this revision, we have made efforts to address this concern by adding more cross-references and highlighting relevant content or formulations. We hope this will make it easier for readers to locate the information they need.
>
> Regarding the calculation of the KL divergence in Equation 23, it may appear abstract and difficult to follow due to a lack of reference to the Clustering Module section (Appendix B.4). We have included cross-references to help readers follow the calculation.
> We appreciate your valuable advice for improving this paper.
>
> ***W5. No solid theoretical guarantee is provided for the proposed method.***
>
> **R5:** We appreciate the reviewer's suggestion and acknowledge the importance of theoretical analysis about extracting or balancing consensus and complementary information in multi-view representation learning. Although we have some promising ideas, we have left them for future works due to the already extensive length and complexity of the paper.
>
> **Reference:**
> - [1]. Chengliang Liu, Jie Wen, Zhihao Wu, Xiaoling Luo, Chao Huang, and Yong Xu. Information recovery-driven deep incomplete multiview clustering network. IEEE Transactions on Neural Networks and Learning Systems, pp. 1–11, 2023. doi: 10.1109/TNNLS.2023.3286918.

---

> ### Author Response · Authors · 2023-11-20
> **Response to Reviewer umVo #2**
>
> ***Q1. What is the first output of NDE module in the output choice of the proposed NDE?***
>
> **A1**: In the Transformer Encoder, which is a sequence-to-sequence model, a sequence of vectors is provided as input and a corresponding sequence of vectors is generated as output. So the "first output" of the NDE module refers to the initial vector in the output vector sequence. It has a bias on the first vector in the input sequence, which is always the most reliable sample in KNN. To provide a more intuitive explanation of the output choice in the NDE and VDE, we have added Figure 5 in the revised manuscript. Additionally, we have updated equations 7 and 14 to incorporate the output choice procedure in the formulation.
>
> ***Q2. How does the Siamese Encoder work in Figure 2? Do you mean the architectures of the upper and lower parts are identical with shared parameters?***
>
> **A2**: Yes, the Siamese Encoder in Figure 2 utilizes identical architectures with shared weights for both the upper and lower parts. This design allows for the extraction of unified representations for both augmented and unaugmented input data. By further incorporating a robustness loss, the Encoder is trained to extract robust representations in the presence of augmentation, and also in the presence of real missing conditions.
>
> ***Q3. How to determine the level of noise added to the original incomplete multi-view data?***
>
> **A3**: In our approach, we employ three types of noise for data augmentation: (1) view dropout, (2) Gaussian noise, and (3) random dropout. For Gaussian noise and random dropout, we conducted a preliminary grid search to determine the optimal noise level. Since the input is normalized, we found that a fixed noise level of 0.05 yields good results for various datasets.
> Regarding view dropout, we conducted an ablation study (Table 14) to determine the best noise level based on different missing rates. We designed Equation 25, which maps the missing rate to the desired noise level. This allows us to dynamically adjust the noise level depending on the missing rate of data.
>
> The results from Table 14 are as follows:
> | Parameter    | $\phi_1 = 0$    | $\phi_1 = 0.15$              | $\phi_1 = 0.3$               | $\phi_1 = 0.45$      | $\phi_1 = 0.6$               | Equation 25   |
> | -------------- | ----------------- | ------------------------------ | ------------------------------ | ---------------------- | ------------------------------ | --------------- |
> | $m_r$ = 0.00 | 89.30 $\pm$ 1.86 | $\underline{89.50 \pm 1.77}$ | 88.91 $\pm$ 1.89              | **89.71 $\pm$ 1.19** | 86.94 $\pm$ 2.12              | $\phi_1=0.15$ |
> | $m_r$ = 0.25 | 86.15 $\pm$ 2.42 | $\underline{88.12 \pm 1.90}$ | **88.56 $\pm$ 1.55**         | 86.51 $\pm$ 3.33      | 83.57 $\pm$ 4.74              | $\phi_1=0.17$ |
> | $m_r$ = 0.50 | 83.35 $\pm$ 1.94 | **86.12 $\pm$ 1.57**         | $\underline{85.61 \pm 2.08}$ | 83.25 $\pm$ 3.86      | 82.25 $\pm$ 4.82              | $\phi_1=0.23$ |
> | $m_r$ = 0.75 | 81.41 $\pm$ 2.79 | $\underline{81.78 \pm 3.86}$ | **82.89 $\pm$ 4.51**         | 80.82 $\pm$ 3.58      | 77.17 $\pm$ 5.63              | $\phi_1=0.32$ |
> | $m_r$ = 1.00 | 76.31 $\pm$ 3.00 | 77.53 $\pm$ 3.63              | 77.02 $\pm$ 3.91              | **80.21 $\pm$ 2.94** | $\underline{79.19 \pm 3.67}$ | $\phi_1=0.46$ |
>
> ***Q4. Do you consider the private information in each view? In addition to the consensus information, the complementary information is also essential, which has been indicated in this paper. However, I cannot understand what is the mechanism used to leverage the view-specific information in the proposed method. Please clarify this in details.***
>
> **A4:** Our proposed method incorporates the preservation of complementary information by training an auto-encoder to reconstruct the input from the unified embedding. This design ensures that the unified embedding have to capture complete information from each view, in order to correctly reconstruction of the multi-view input. Consequently, both consensus and complementary information are retained within the unified embedding. Furthermore, since the unified embedding is directly used for clustering, both types of information are effectively leveraged in our approach.
>
> For a more detailed discussion and evidence regarding the preservation of complementary information, we kindly refer you to our response R1 to Reviewer LBwN, where we provide additional insights and supporting evidence.

---

> > ### Comment · Reviewer_umVo · 2023-11-22
> >
> > Thanks for your response. My final decision will be released soon, after the discussion with other reviewer and AC.

---

> > > ### Author Response · Authors · 2023-11-22
> > >
> > > Thanks again for your valuable feedback and suggestions to improve this paper.

---

### Official Review · Reviewer_3U96 · 2023-10-30

**Soundness:** 2 fair
**Presentation:** 2 fair
**Contribution:** 2 fair
**Rating:** 5
**Confidence:** 5

**Summary:**

In this paper, a deep representation learning network is proposed for incomplete multi-view clustering. The method exploits the KNN imputation approach to fill the missing views and integrates the augmentation strategy. Many experiments, especially many ablation experiments are conducted to validate the method.

**Strengths:**

The authors conducted many ablation experiments to validate the methods.

**Weaknesses:**

1. The experiments are not sufficient. Firstly, there are no experiments on large-scale datasets. Secondly, the authors only evaluate the method on the image datasets, where all views are extracted from the image.
2. Efficiency and computational complexity are also the very important metric to evaluate the method. However, these are ignored.
3. The novelty of the method seems not strong but the method seems very complex. Imputation the missing views for incomplete multi-view clustering is not new and has many related works. For example, the work ‘Deep safe incomplete multi-view clustering: Theorem and algorithm’ also exploits the KNN imputation for missing views. The method used Augmentation and KNN imputation to fill the missing views. However, the authors do not visualize the imputed missing views. This is not reasonable. In many existing works, such as ‘Dual contrastive prediction for incomplete multi-view representation learning’, the imputed missing views can be visualized to make the approach look more credible. However, just using ablation experiments is not convincing enough.

**Questions:**

1. How to validate the robustness as a robust method proposed in the paper?
2. K-Nearest-Neighbor (KNN) imputation is introduced in the paper. Is the method sensitive to the nearest neighbor numbers?
3. From Table 1, the feature dimensions of the datasets are not large, even very small. For example, one feature dimension of handwritten datasets is just 6. Is it necessary to use deep neural networks even Transformer to extract its features again?
4. What is the impact of the design of deep neural network layers and the selection of dimensions for each layer on clustering results?
5. For the experimental results, why are the experimental results you provided lower than the original papers? For example, DCP on the Scene 15 dataset is much lower than the published papers. In addition, how were the experimental results of Completer obtained? The original Completer is proposed for two view data which cannot be applied on the datasets you exploited in the paper directly.

---

> ### Author Response · Authors · 2023-11-20
> **Response to Reviewer 3U96 #1**
>
> ***W1: The experiments are not sufficient. Firstly, there are no experiments on large-scale datasets. Secondly, the authors only evaluate the method on the image datasets, where all views are extracted from the image.***
>
> **R1:** We would like to address the concerns regarding the experiments.
>
> Firstly, we would like to note that the largest datasets we used in our experiments are Reuters (18,758 samples) and ALOI_Deep (10,800 samples), which are roughly at the same scale as prior works such as RecFormer [1] (10,800 samples), SURE [2] (30,000 samples), and DSIMVC [3] (6,773 samples). We acknowledge that experiments on larger-scale datasets are important and we plan to explore them in future work due to limited time.
>
> Secondly, we acknowledge the limitation of only evaluating our method on image datasets where all views are extracted from the image. To address this concern, we have made revisions to our manuscript. In this revision, we have included two textual datasets for comparison. The statistics of these datasets are summarized in Table 4, and the results of our approach with textual features are recorded in Table 5 in Appendix C.3. BDGP is a multi-modal dataset with 1,750-dimensional visual features and 79-dimensional textual features as two views, while Reuters is a multilingual dataset where English and French are used as two views. The results achieved by our approach with textual features demonstrate its competitive effectiveness in different modalities. The main results of Table 5 are as follows:
>
>
> | Dataset\Metrics   | BDGP Acc(\%)                 | BDGP NMI(\%)                 | BDGP ARI(\%)                 | Reuters Acc(\%)              | Reuters NMI(\%)              | Reuters ARI(\%)              |
> | ------------------ | ------------------------------ | ------------------------------ | ------------------------------ | ------------------------------ | ------------------------------ | ------------------------------ |
> | Completer        | 58.37 $\pm$ 4.87 | 48.52 $\pm$ 4.34 | 25.13 $\pm$ 6.71 | 40.30 $\pm$ 7.32 | 22.86 $\pm$ 6.44 | 10.53 $\pm$ 8.25 |
> | DSIMVC           | **95.71 $\pm$ 0.21**         | **87.08 $\pm$ 0.54**         | **89.62 $\pm$ 0.51**         | 48.39 $\pm$ 2.92 | **31.88 $\pm$ 2.27**         | **26.04 $\pm$ 2.27**         |
> | SURE             | 63.27 $\pm$ 7.55 | 41.35 $\pm$ 7.70 | 36.51 $\pm$ 8.87 | $\underline{48.63 \pm 3.56}$ | 27.73 $\pm$ 2.46 | 22.46 $\pm$ 1.59 |
> | DCP              | 55.82 $\pm$ 7.02 | 44.56 $\pm$ 8.92 | 20.08 $\pm$ 11.71 | 39.02 $\pm$ 3.16 | 22.47 $\pm$ 4.19 | 6.92 $\pm$ 3.34$              |
> | CPSPAN           | 81.40 $\pm$ 8.64 | 66.19 $\pm$ 6.61 | 64.71 $\pm$ 9.89 | 39.35 $\pm$ 2.13 | 14.47 $\pm$ 2.01 | 12.37 $\pm$ 1.79 |
> | RecFormer        | 49.76 $\pm$ 3.49 | 38.62 $\pm$ 3.30 | 19.01 $\pm$ 2.68 | 39.70 $\pm$ 5.14 | 17.27 $\pm$ 2.83 | 14.82 $\pm$ 3.19 |
> | URRL-IMVC (ours) | $\underline{92.52 \pm 5.71}$ | $\underline{82.29 \pm 5.66}$ | $\underline{84.26 \pm 7.57}$ | **49.91 $\pm$ 2.44** | $\underline{29.49 \pm 1.38}$ | $\underline{25.71 \pm 2.08}$ |
>
> ***W2: Efficiency and computational complexity are also the very important metric to evaluate the method. However, these are ignored.***
>
> **R2**: Thank you for bringing this to our attention. We have now included an analysis of model parameters and computational costs (Multiply-Adds, MACs) in Table 8 of Appendix C.3. The results demonstrate that our model surpasses other state-of-the-art DIMVC methods in terms of both model size and computational cost. It is worth mentioning that our network hyperparameters, such as the dimensions of the feed-forward networks, were roughly chosen similarly to prior DIMVC methods, and we maintained the same configuration (except for the input layers) across all datasets. However, we acknowledge that there is still room for improvement in carefully designing the network architecture tailored to different datasets. The main results of Table 8 are as follows:
>
>
> | Approach         | MACs (M)           | Parameters (M)     |
> | ------------------ | -------------------- | -------------------- |
> | Completer        | 8.98               | 6.82               |
> | DSIMVC           | 9.89               | 9.33               |
> | SURE             | 8.24               | 8.24               |
> | DCP              | 9.02               | 9.01               |
> | CPSPAN           | $\underline{4.92}$ | $\underline{4.93}$ |
> | RecFormer        | 12.08              | 6.09               |
> | URRL-IMVC (ours) | **1.54**           | **1.01**           |

---

> ### Author Response · Authors · 2023-11-20
> **Response to Reviewer 3U96 #2**
>
> ***W3: The novelty of the method seems not strong but the method seems very complex. Imputation the missing views for incomplete multi-view clustering is not new and has many related works. For example, the work ‘Deep safe incomplete multi-view clustering: Theorem and algorithm’ also exploits the KNN imputation for missing views. The method used Augmentation and KNN imputation to fill the missing views. However, the authors do not visualize the imputed missing views. This is not reasonable. In many existing works, such as ‘Dual contrastive prediction for incomplete multi-view representation learning’, the imputed missing views can be visualized to make the approach look more credible. However, just using ablation experiments is not convincing enough.***
>
> **R3:** We appreciate the reviewer's comments and we would like to clarify certain points. Although KNN Imputation is widely used in prior IMVC approaches, its utilization in our framework and the way to further process the imputation is different.
>
> In contrast to previous approaches like DSIMVC [3] and CPSPAN [4], where KNN Imputation is used as a missing view recovery strategy in the latent space, our framework is recovery-free, as mentioned in the paper (page 3, contribution 2). KNN Imputation serves as a preprocessing strategy in the data space, providing hints for learning a robust representation. **While the KNN Imputation algorithm itself is not novel, its utilization in our framework is different**.
>
> Additionally, unlike prior works that use naive strategies to fuse searched KNNs, we retain the noisy raw KNN data and fuse it with well-designed attention-based Encoders (Neighbor Dimensional Encoder, NDE). **This approach is distinctive and explains why we did not visualize the imputed missing views, as they are raw data from the dataset**.
>
> **Finally, KNN Imputation is just a component of our novel framework**, which directly learns a unified and robust representation for the IMVC task, surpassing the limitations posed by existing approaches in the field.
>
> **Unified**: In the DIMVC field, the prevailing framework is cross-view contrastive learning. However, this design primarily emphasizes the consensus information in multi-view data while neglecting the valuable complementary information. Therefore, our approach goes beyond the cross-view contrastive framework and instead learns a unified representation using a meticulously designed attention-based network that effectively leverages complementary information.
>
> **Robust**: The primary approach for handling missing views in the DIMVC field is to first recover the missing views and then perform complete multi-view clustering. However, the recovered views may be unreliable, and computational resources are wasted on repeatedly mapping between data and latent space [1]. To address this issue, we propose directly learning a latent embedding that is robust to view missing, without explicitly recovering the missing data. We have devised data augmentation and KNN imputation as preprocessing techniques, which, along with the unified representation learning framework, facilitate efficient and robust representation learning.
>
>
> **Reference:**
> - [1]. Chengliang Liu, Jie Wen, Zhihao Wu, Xiaoling Luo, Chao Huang, and Yong Xu. Information recovery-driven deep incomplete multiview clustering network. IEEE Transactions on Neural Networks and Learning Systems, pp. 1–11, 2023. doi: 10.1109/TNNLS.2023.3286918.
> - [2]. Mouxing Yang, Yunfan Li, Peng Hu, Jinfeng Bai, Jiancheng Lv, and Xi Peng. Robust multi-view clustering with incomplete information. IEEE Transactions on Pattern Analysis and Machine Intelligence, 45(1):1055–1069, 2023b. doi: 10.1109/TPAMI.2022.3155499.
> - [3]. Huayi Tang and Yong Liu. Deep safe incomplete multi-view clustering: Theorem and algorithm. In Kamalika Chaudhuri, Stefanie Jegelka, Le Song, Csaba Szepesvari, Gang Niu, and Sivan Sabato (eds.), Proceedings of the 39th International Conference on Machine Learning, volume 162 of Proceedings of Machine Learning Research, pp. 21090–21110. PMLR, 17–23 Jul 2022.
> - [4]. Jiaqi Jin, Siwei Wang, Zhibin Dong, Xinwang Liu, and En Zhu. Deep incomplete multi-view clustering with cross-view partial sample and prototype alignment. In Proceedings of the IEEE/CVF Conference on Computer Vision and Pattern Recognition (CVPR), pp. 11600–11609, June 2023.

---

> ### Author Response · Authors · 2023-11-20
> **Response to Reviewer 3U96 #3**
>
> ***Q1: How to validate the robustness as a robust method proposed in the paper?***
>
> **A1:** We believe that the robustness of our proposed method can be explained from three aspects.
>
> Firstly, its stability has been validated through multiple runs of the same experiment. Table 2 presents the results of 10 runs, showing that our approach achieved a relatively low standard deviation. This indicates the reliability and consistency of our method.
>
> Secondly, the robustness of our method has been examined under different view-missing conditions. Figure 3 demonstrates that our approach exhibits a smaller decline in performance compared to other state-of-the-art methods even when the missing rate reaches 0.75. This suggests that our method is more robust in handling missing data.
>
> Finally, the robustness of our approach is determined by experiments on various datasets, including image-based and text-based. Our approach consistently achieves state-of-the-art performance, proving its robustness to different data types.
>
> It is also worth noting that when we use the term "robust" in the title, we specifically refer to the robustness of the embeddings generated by our method. This refers to the ability of the unified embedding to remain consistent even under different missing conditions. To ensure this robustness, we have incorporated a robustness loss and an augmentation strategy during the training process. These mechanisms work together to enhance the stability and consistency of the embeddings, ultimately resulting in a more robust method.
>
> ***Q2: K-Nearest-Neighbor (KNN) imputation is introduced in the paper. Is the method sensitive to the nearest neighbor numbers?***
> **A2:** According to current observations, near-optimal results can be achieved with k>=4. The related Ablation study has been presented in Table 15. The result is unimodal with the best k = 4, and larger k values tend to provide more stable results (smaller standard deviation). The main results of Table 15 are as follows:
>
> | k=1 | k=2 | k=4 | k=8 | k=16 |
> | --- | --- | --- | --- | --- |
> | 83.84 $\pm$ 3.38 | 84.84 $\pm$ 2.82 | **87.31 $\pm$ 2.01** | $\underline{87.00 \pm 2.46}$ | 86.01 $\pm$ 1.28 |
>
> ***Q3: From Table 1, the feature dimensions of the datasets are not large, even very small. For example, one feature dimension of handwritten datasets is just 6. Is it necessary to use deep neural networks even Transformer to extract its features again?***
>
> **A3:** It is important to consider the use of deep neural networks, or even Transformers, to extract features again for datasets with small feature dimensions, such as the handwritten datasets mentioned in Table 1. While it may seem unnecessary at first glance, there are several reasons why we believe it can be beneficial for the IMVC task.
>
> Firstly, raw data often lack sufficient clustering characteristics. For instance, in the case of the Handwritten dataset, simply concatenating the data and performing clustering does not lead to satisfactory results, as demonstrated in RecFormer [1]. This highlights the need for additional feature extraction methods to capture more meaningful representations.
>
> Secondly, deep neural networks and Transformers not only extract features but also possess the capability to store learned knowledge, such as data correlation and distribution. These learned representations are crucial for handling complex cross-view correlations and missing data conditions.
>
> Finally, in our specific case, Transformers are employed to fuse information from multiple views and incorporate knowledge from nearest neighbors.

---

> ### Author Response · Authors · 2023-11-20
> **Response to Reviewer 3U96 #4**
>
> ***Q4: What is the impact of the design of deep neural network layers and the selection of dimensions for each layer on clustering results?***
>
> **A4:** We performed an ablation study to investigate the impact of the design of deep neural network layers and the selection of dimensions. The results of this study are presented in Table 13 in Appendix C.6.
>
> Our findings suggest that larger models do not necessarily lead to better clustering accuracy. In fact, the best results were obtained with a balanced setting, where there were 2 Transformer blocks in both the NDE and VDE, and the FFN and embedding dimensions were set to 256.
>
> This observation can be explained by the fact that small models may struggle to learn the complex fusion mapping between raw data and unified embedding, while larger models can easily overfit and weaken the relationship between the latent space and data space.
>
> It is worth noting that the design of deep neural network layers and the selection of dimensions have not been a primary focus in prior works. For example, some previous studies have used simplistic designs such as 3-layer MLPs. Therefore, the network hyperparameters in our approach were not finely tuned, and the same setting was applied to all datasets.
>
> The main results of Table 16 are as follows:
> | Transformer Blocks | FFN Dimension | Embedding Dimension | MACs | Parameters | Accuracy |
> | ------------------ | ------------- | --------------------- | ---- | ---------- | -------- |
> | 1 | 128 | 128 | 29.73 | 12.07 | 91.86 $\pm$ 1.80 |
> | 2 | 128 | 128 | 33.65 | 13.03 | 91.29 $\pm$ 3.24 |
> | 2 | 256 | 128 | 43.04 | 16.69 | 90.33 $\pm$ 5.26 |
> | 2 | 256 | 256 | 43.98 | 17.13 | **92.95 $\pm$ 2.60** |
> | 4 | 256 | 256 | 60.08 | 21.07 | $\underline{91.92 \pm 2.57}$ |
> | 4 | 512 | 256 | 97.05 | 34.14 | 89.31 $\pm$ 5.11 |
> | 4 | 512 | 512 | 103.34 | 36.39 | 90.02 $\pm$ 3.29 |
>
> ***Q5: For the experimental results, why are the experimental results you provided lower than the original papers? For example, DCP on the Scene 15 dataset is much lower than the published papers. In addition, how were the experimental results of Completer obtained? The original Completer is proposed for two-view data which cannot be applied to the datasets you exploited in the paper directly.***
>
> **A5:** It is common to observe differences in experimental results compared to original papers due to various factors such as dataset settings, view missing conditions, randomness, and adapting strategies. For example, the Handwritten dataset used in RecFormer [1] has 5 views, while CPSPAN [2] used a variant with 6 views. There is also a discrepancy in the number of samples in the Caltech7 dataset reported by these two methods (1474 vs 1400), and their reported results have a 35% difference in accuracy (86% vs 51%) on this dataset.
>
> To ensure a fair comparison, we aligned the results of prior works with our dataset settings as described in Section 4.1 and Appendix C.2. For all methods compared in Table 2, we reproduced their results on our chosen dataset and used the same missing conditions. In Table 4, we adapted our method to match the settings in MPC [3]. However, some prior works are challenging to adapt to different numbers of views, which can hinder real-world applications. For example, the Completer method can only handle 2-view datasets, and the DCP method requires extensive modifications to handle datasets with more than 3 views. In such cases, we randomly select views to evaluate on datasets with a higher number of views. We have provided detailed experimental information in Appendix C.2 for a clearer understanding of our experimental setup.
>
> The reported lower result for DCP compared to the original paper is due to the difference in the number of views. Our Scene-15 dataset has only 2 views, whereas the DCP paper considered 3 views. Previously, an incorrect adaptation from the 3-view implementation to the 2-view dataset was used. We have now reproduced the result using the correct setup and updated it in Table 2. The result now is slightly lower than the original paper, which may be due to the randomness or the difference in the number of views.
>
> For Completer, we randomly select 2 views for evaluation on datasets with more than 2 views since the method is challenging to adapt to a higher number of views.
>
> Besides, we have carefully checked the experimental settings of reproduced methods and found that the experimental results are consistent with the original papers. We hope this revised response addresses your concerns. Thank you for your valuable feedback.

---

> ### Author Response · Authors · 2023-11-20
> **Response to Reviewer 3U96 #5**
>
> **Reference:**
> - [1]. Chengliang Liu, Jie Wen, Zhihao Wu, Xiaoling Luo, Chao Huang, and Yong Xu. Information recovery-driven deep incomplete multiview clustering network. IEEE Transactions on Neural Networks and Learning Systems, pp. 1–11, 2023. doi: 10.1109/TNNLS.2023.3286918.
> - [2]. Jiaqi Jin, Siwei Wang, Zhibin Dong, Xinwang Liu, and En Zhu. Deep incomplete multi-view clus- tering with cross-view partial sample and prototype alignment. In Proceedings of the IEEE/CVF Conference on Computer Vision and Pattern Recognition (CVPR), pp. 11600–11609, June 2023.
> - [3]. Junjie Liu, Junlong Liu, Shaotian Yan, Rongxin Jiang, Xiang Tian, Boxuan Gu, Yaowu Chen, Chen Shen, and Jianqiang Huang. Mpc: Multi-view probabilistic clustering. In Proceedings of the IEEE/CVF Conference on Computer Vision and Pattern Recognition (CVPR), pp. 9509–9518, June 2022.

---

### Author Response · Authors · 2023-11-20
**General Response by Authors**

## Summarization of Revisions & Common Concerns
We appreciate the reviewer's thoughtful feedback and try our best to revise the manuscript and address all concerns.

### Common Concerns

***Q1: Concerns about Experiments***

**A1:** We notice that there are concerns about experiments in our paper. Due to the 9-page paper length limitation of this conference, most of the ablation studies and several other experiments were included in Appendix C.5 and C.6. This might have caused them to be easily overlooked by readers. To address this issue, we have revised our paper to emphasize the cross-reference to the relevant sections in the appendix. We have also conducted additional experiments as recommended by the reviewers, and the results have been included in the revised manuscript to provide evidence or comparison with SOTA methods. Please refer to the Main Revisions section for more details.

To summarize, we compared our approach with 11 state-of-the-art approaches on 8 datasets with textual and image features. We conducted 9 ablation studies to determine the effectiveness of our design and hyperparameters, and 2 visualization experiments are included to provide explanations of the training process.

***Q2: Concerns about detailed explanation or formulation of modules.***

**A2:** We recognize the concern regarding the detailed explanation and formulation of modules in our paper. Similarly, we had to move most of these details to Appendix B to comply with the paper length limitation. However, we understand that it might be challenging for readers to locate these details. To address this issue, we have added more cross-references throughout the paper to facilitate easy access to the relevant content in the appendix. Additionally, for frequently asked details such as the KNN Imputation algorithm, we have provided more explanations and details to ensure clarity and understanding.

### Summarization of Revisions

1. ***Add Table 4 and 5***: Provide comparison with state-of-the-art methods on textual datasets.
2. ***Add Table 7***: Provide comparison with cross-view contrastive learning-based approaches under different numbers of views.
3. ***Add Table 8***: Compare computational cost and model parameters with state-of-the-art methods.
4. ***Add Table 12***: Study the effect of using the missing view recovery strategy.
5. ***Add Table 16***: Ablation study about network hyperparameters.
6. ***Add Figure 5***: An intuitive visualization of output choice of Neighbor Dimensional Encoder and View Dimensional Encoder
7. ***Add Figure 6***: Visualize loss and clustering performance throughout the training process.
8. ***Add Algorithm 1***: Provide a more detailed explanation of the KNN Imputation algorithm.
9. ***Updated Appendix C.2***: More explanations and details about our experimental setup are included.
10. ***Updated equations***: Several equations including equations 7 and 14 are updated for correctness or completeness of the formulation.
11. ***Add and emphasize cross reference***: Facilitate readers find relative contents.

---

### Meta-Review · Area_Chair_Ue3Y · 2023-12-01

**Metareview:**

This paper worked on multi-view clustering and proposed a unified and robust representation learning method without using cross-view contrastive learning or missing view recovery. Perhaps there is no major issue --- just no reviewer became excited, because its novelty and significance are overall not competitive enough. As a result, our reviewers consistently rated it borderline reject and agreed to reject it.

**Justification For Why Not Higher Score:**

Its novelty and significance are overall not competitive enough.

**Justification For Why Not Lower Score:**

N/A

---

### Decision · Program_Chairs · 2024-01-16

Reject